# Multistep orthophosphate release tunes actomyosin energy transduction

Luisa Moretto[1,3], Marko Ušaj [1,3], Oleg Matusovsky [2,3], Dilson E. Rassier [2]✉, Ran Friedman [1]✉ & Alf Månsson [1]✉

Muscle contraction and a range of critical cellular functions rely on force-producing interactions between myosin motors and actin filaments, powered by turnover of adenosine triphosphate (ATP). The relationship between release of the ATP hydrolysis product ortophosphate (Pi) from the myosin active site and the force-generating structural change, the power-stroke, remains enigmatic despite its central role in energy transduction. Here, we present a model with multistep Pi-release that unifies current conflicting views while also revealing additional complexities of potential functional importance. The model is based on our evidence from kinetics, molecular modelling and single molecule fluorescence studies of Pi binding outside the active site. It is also consistent with high-speed atomic force microscopy movies of single myosin II molecules without Pi at the active site, showing consecutive snapshots of pre- and post-power stroke conformations. In addition to revealing critical features of energy transduction by actomyosin, the results suggest enzymatic mechanisms of potentially general relevance.

Utilization of the chemical free energy of ATP (We use the abbreviation ATP and ADP while recognizing that MgATP and MgADP are the actual substrate and product, respectively.) for force- and motion-generation by myosin molecules in their interaction with actin filaments, is a prime example of chemomechanical energy transduction in biology. The process underlies critical cell and body functions[1–3] in animals and plants from muscle contraction over non-muscle cell motility and intracellular transport to cell signaling. It is well-known that Pi-release from the myosin active site is a central step in the energy transduction[4–25]. However, it is controversial whether the main force-generating structural change, the powerstroke, gates Pi-release or vice versa[7,10,12,15,16,21,24,26–29], a lack of insight that hampers understanding of the transduction process.

Experiments that combine transient biochemical kinetics with time-resolved fluorescence resonance energy transfer (FRET) for powerstroke detection provide particularly compelling evidence that Pi-release occurs after the powerstroke[7,8]. These findings are critical to take into account in a model relating Pi-release to force generation.

The idea that Pi-release from the active site occurs after the power-stroke also fits with mechanisms of the basal myosin ATPase[22], transient changes in tension in response to altered [Pi] (Pi-transients)[9] and [Pi]-independent rates of fast mechanical transients in muscle fibers and single molecules[6,10,27]. In contrast, other mechanical experiments on isolated proteins[30], along with X-ray crystallography, mutagenesis, and related biochemical kinetics data[2,16], suggest that Pi-release from the active site precedes the powerstroke. The latter idea, with the gating of the powerstroke by Pi-release, is also supported by theoretical considerations[28]. According to the latter, Pi-release after the powerstroke would be associated with reduced muscle shortening velocity and only a minimal change in isometric force upon increased [Pi], in stark contrast to experimental findings[5,13,23,31]. Finally, a slow Pi-release after the power-stroke, of rate $<120\,s^{-1}$, as observed experimentally[7,16,32], is difficult to reconcile with the high shortening velocity of muscle. The latter is instead generally believed to be rate-limited by the ADP-release and ATP-induced detachment rates $>1000\,s^{-1}$[33]. To summarize, there is strong evidence from transient

[1]Department of Chemistry and Biomedical Sciences, Linnaeus University, SE-391 82 Kalmar, Sweden. [2]Department of Kinesiology and Physical Education, McGill University, Montreal, QC H2W 1S4, Canada. [3]These authors contributed equally: Luisa Moretto, Marko Ušaj, Oleg Matusovsky. ✉e-mail: dilson.rassier@mcgill.ca; ran.friedman@lnu.se; alf.mansson@lnu.se

kinetics studies[7,16,32] that Pi from the active site reaches the bulk solution with a delay after the power-stroke. This idea could, most simplistically, be fit by a model with Pi-release after the power-stroke. Whereas such a model is consistent with the response of muscle and isolated proteins to transient perturbations it seems impossible to reconcile with other findings, such as the high shortening velocity of muscle and its lack of [Pi] dependence. The latter results are more readily accounted for by a model with Pi-release before the power-stroke. The above conflicting findings pose severe challenges

for a full understanding of the mechanism relating Pi-release to the power-stroke.

A potential to resolve at least some of the mentioned controversies emerged[2,16] when structural studies of myosin VI[16] suggested that Pi pauses at a secondary binding site in the Pi-release tunnel (Fig. 1a) on its way from the active site to bulk solution. Such a phenomenon could explain the delay of Pi-appearance in the solution until after the power-stroke even if Pi leaves the active site before the stroke. However, myosin II[7,34] and myosin V[8,27], not myosin VI, have been used

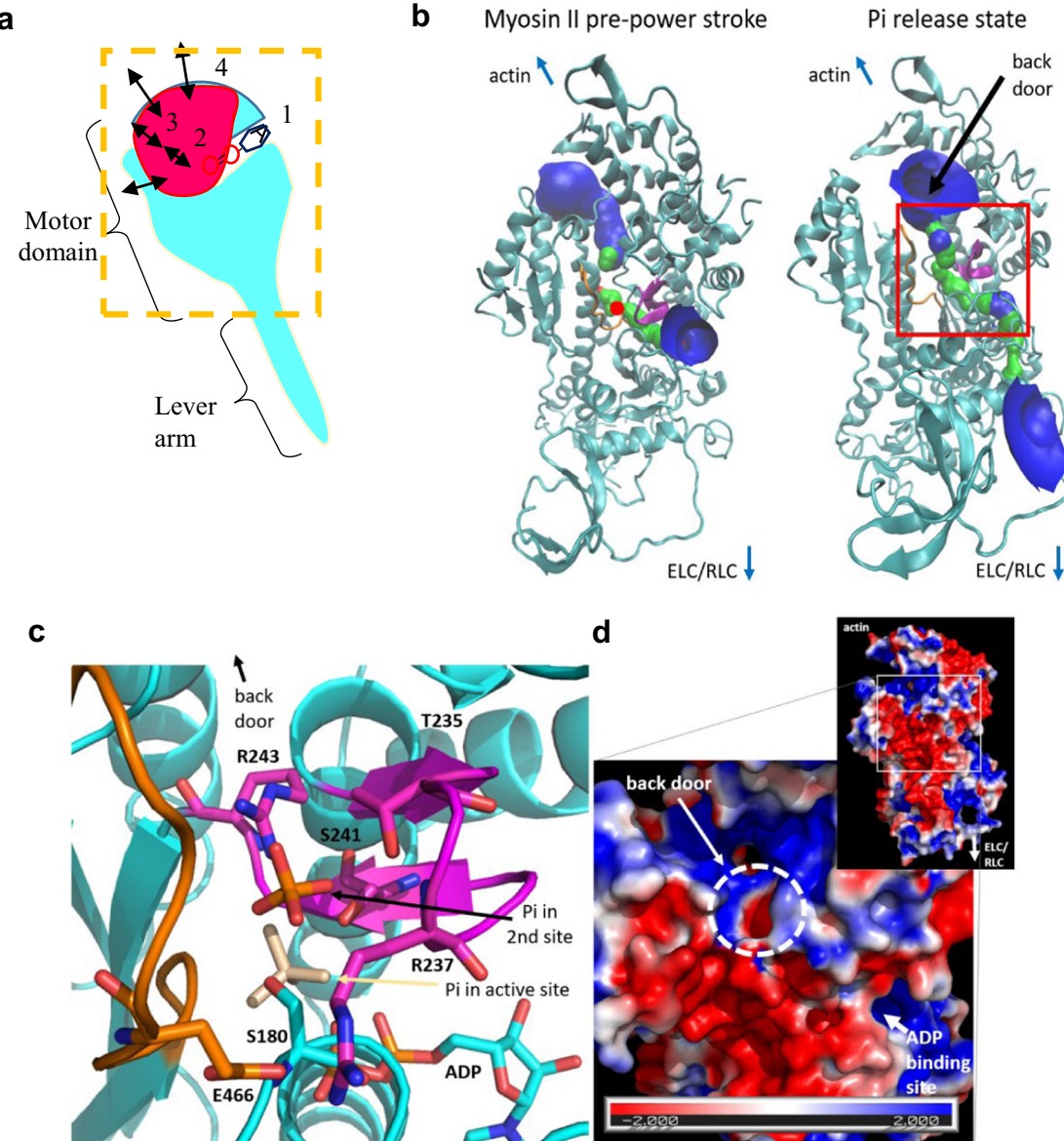

**Fig. 1 | Modeling of Pi-release of striated muscle (cardiac) myosin II. a** Schematic illustration of Pi-release from myosin subfragment 1: with binding of ATP (1), Pi-binding to the active site (2) Pi-binding to the secondary site (3) release via the back door and finally weak electrostatic binding to surface sites (4) on the myosin head. The location of Pi in well-defined positions is indicated by solid spheres. Arrows indicate the routes of Pi from ATP binding to release via the back door and its weak binding on the motor surface. The orange dashed box indicates part of subfragment 1 shown in (**b**). **b** Structure of the myosin II motor domain in the pre-powerstroke state (left; PDB code 5N6A (Pre-powerstroke)) and molecular model (right) of a Pi-release state generated from the 5N6A (Pre-powerstroke)structure by using the switch II conformation from myosin VI (PDB 4PFO (Pi-release state)). Analysis of the available space in the structure (using the HOLE computer program) indicated that,

in the pre-powerstroke state, the Pi-release path via the back door is closed (red patches). In contrast, this path is fully open in the modeled Pi-release state (green and blue; see Methods). The red box indicates the region shown in detail in **c**. Actin binding site is indicated as well as the direction (arrow) where the essential (ELC) and regulatory light chain (RLC) are found. **c** Structural model of myosin II in our modeled Pi-release state (**b**; right, red box) with phosphate in the secondary site (orange). The position of Pi in the active site in the pre-powerstroke state is also indicated (beige). To allow the release of the phosphate, Switch II (orange wire) has moved from its initial pre-powerstroke state to the modeled state shown. Switch I shown in purple. **d** Electrostatics around the opening of the back door in myosin II (PDB: 5N6A (Pre-powerstroke)). The surface is colored according to its surface electrostatic potential, from −2.0 $k_B$T (deep red) to +2.0 $k_B$T (deep blue).

in the biophysical/biochemical experiments that relate the power-stroke to Pi-release. It is therefore important to clarify whether Pi-binding in the Pi-release tunnel outside the active site, i.e., similar to that observed in myosin VI[2,16], may occur also in myosin II. Another possibility is additional Pi-binding sites outside the Pi-release tunnel, on the surface of the myosin motor domain. This idea is consistent with recent observations[35] suggesting nonspecific binding of fluorescent ATP to myosin motor domains, effects that may partly be mediated via the phosphate groups of ATP. However, to the best of our knowledge, such Pi-binding sites have not been explicitly demonstrated, and, if they exist, their contribution to a delayed Pi-appearance in solution is unclear. Finally, we hypothesize that if Pi-binding site(s) exist outside the active site, this could also reconcile the above-mentioned conflicting experimental observations related to the power-stroke and Pi release[7,10,12,16,21,24,26–28]. These include the [Pi]-dependence of isometric force and the simultaneous [Pi]-independence of power-stroke rate and shortening velocity.

Here, we address the outstanding issues through multiple approaches. First, using molecular modeling and a single molecule competition assay, we present evidence for Pi-binding to more than one site outside the active site in myosin II. Next, we incorporate this feature into a multi scale mechanokinetic model (developed from ref. [20]), allowing rigorous analysis of the effects of secondary Pi-binding on contractile function. Strikingly, we find that the model reconciles the effects of altered [Pi] on mechanical transients[6,10,27] and steady-state force/velocity data[5,13,23,31], previously viewed as contradictory. Finally, high-speed atomic force microscopy experiments allow us to verify a critical model prediction that the power-stroke, and its reversal, occurs without Pi, i.e., with ADP only at the myosin active site.

Whereas the study focuses on striated muscle myosin II, our results suggest appreciable similarities with the Pi-release mechanism in other myosins, implying that the mechanism is general over the myosin superfamily. Furthermore, the type of multistep product release found may have roles in the effective operation also of non-motor enzymes.

## Results

### Orthophosphate binding to myosin outside the active site

Due to the absence of relevant molecular structures of myosin II from experiments, we used molecular modeling to investigate if the evidence from studies of myosin VI[16], of secondary Pi-binding in the release tunnel (the back door) between the active site and bulk solution, can be extrapolated to myosin II. First, in analogy to the situation in myosin VI, the analysis suggests (Fig. 1b, left) that the back door is not available in myosin II in the pre-powerstroke state.

The results of this analysis are in agreement with previous simulation studies where enhanced sampling was used to suggest possible pathways for Pi-release[36,37]. Given that switch II must move to allow phosphate exit via the back door pathway, we introduced the switch II structure of the Pi-release state of myosin VI into the structure of bovine cardiac myosin II (ventricular β-myosin, MYH7 gene). Testing the resulting model (Fig. 1b, right) indicates the availability of the back door, like in myosin VI (Supplementary Fig. 1). Indeed, in a recent study of myosin VI[37], it has been suggested (by using multiple long molecular dynamics simulations) that the back door route is dominant, even if other routes might also be possible. Noticeably, a similar back door structure can be identified in Dictyostelium myosin II modeled (Supplementary Fig. 1b, right) from a crystal structure of this myosin[16] (after removal of mutations in the experimental data). Further, the resulting molecular model indicates that cardiac myosin II in the modeled Pi-release state (Fig. 1c), in analogy to myosin VI[16] (Supplementary Fig. 2), has a secondary Pi-binding site outside the active site where Pi can pause on its way to bulk solution. Quantum mechanical calculations, suggest binding energy of −12 $k_B$T for Pi-binding at the secondary site of myosin II. Despite the possibility that limitations of the model might

render the actual binding energy less favorable than calculated, the data are consistent with tight Pi-binding compatible with the observed slow off-rate of Pi, compared to the power-stroke rate[7,16,32]. Unfortunately, however, the actual off-rate of Pi from this secondary site is currently inaccessible, both to experimental measurements and molecular modeling. In the latter case, there are simply too many complexities that cannot be faithfully included in the modeling, e.g., the exact exit path through the release tunnel, viscosity of the trapped solvent, dynamic modifications to the electrostatic field, etc.

In addition to the secondary site (Fig. 1c), the surface electrostatics (Fig. 1d and Supplementary Fig. 3) of the myosin head at the exit of the back door suggests the possibility of additional electrostatic Pi-binding. This is in accordance with what we hypothesized above based on recent evidence[35] suggesting nonspecific binding of fluorescent ATP to the myosin motor domain. The latter type of Pi-binding should, if existing, be readily accessible to experimental verification in contrast to binding to the secondary site considered above. Thus, we used a competitive assay[38], assessing potential Pi-binding to surface sites of myosin by inhibition of the binding of a fluorescent ATP analog (Alexa647-ATP) detected by total internal reflection fluorescence (TIRF) based single-molecule microscopy (Fig. 2 and Supplementary Movie 1). We first adsorbed the myosin motor fragment, heavy meromyosin (HMM), to a silanized surface and then added Alexa647-ATP in the presence of different concentrations of Pi. The number of observed fluorescent spots decreased as [Pi] increased in the range of 0.1–43 mM (Fig. 2a, b), consistent with one or several binding constants in the range of 0.1–10 mM. Our results showed that both a slow phase (attributed to ATP turnover by myosin) and fast phases of Alexa647-ATP binding to myosin were reduced in amplitude by increasing [Pi] (Fig. 2c, d). Recent experimental results, comparing isolated HMM and myosin subfragment 1 motor domains, convincingly attributed the fast phases to ATP binding to the myosin head outside the active site[35]. It is of interest to note that similar auxiliary ATP binding sites have recently been demonstrated in another motor protein[39]. The reduction in amplitude of the fast phases in Fig. 2, and similar behavior when the active site is blocked by vanadate (Fig. 3), are therefore difficult to explain in any other way than by Pi-binding outside the active site that competes with the Alexa647-ATP binding. Based on binding constants in the 0.1–10 mM range (Fig. 2b), calculations in Supplementary Fig. 4 suggest off-rate constants >150 s⁻¹ for Pi-binding to these sites. This is outside the range found experimentally for Pi-release (∼<120 s⁻¹)[7,16,32]. Nevertheless, the possible range of the off-rate constant suggests that binding to the surface sites may contribute to a delayed appearance of Pi in solution. However, a substantial fraction of the delay remains unaccounted for and we attribute that fraction to binding to the secondary site in Fig. 1c.

Overall, our results are consistent with a multistep Pi-release (Fig. 1a) where Pi, after leaving the active site, is guided along the back door with temporary binding to at least two, and possibly more, sites outside the active site (Figs. 1 and 2). The overall energetics of the process is thus governed by multiple equilibria: $K_{Tot} = K_{Pr}K_{C}'K_{C}''$ where $K_{Pr}$ is the equilibrium constant between the states with Pi at the active site and Pi in the secondary site, $K_C'$ is the equilibrium constant between the latter state and that with Pi attached to the myosin surface. Finally, $K_C''$ is the equilibrium constant for the binding of Pi in solution to the surface sites. The multiple equilibria have important implications. First, the activation energy for Pi-release from the active site is partitioned, with at least two free energy minima on the way from the active site to the bulk solution. Second, the coupled equilibria would allow versatile regulation of both overall Pi-affinity and kinetics by modulating either of the transitions governed by the equilibrium constants $K_{Pr}$, $K_C'$ or $K_C''$. For example, the value of $K_{Pr}$ is changed between the Pi-release state and post-powerstroke states (cf. Fig. 4a). One may also consider differences in $K_C''$ between different

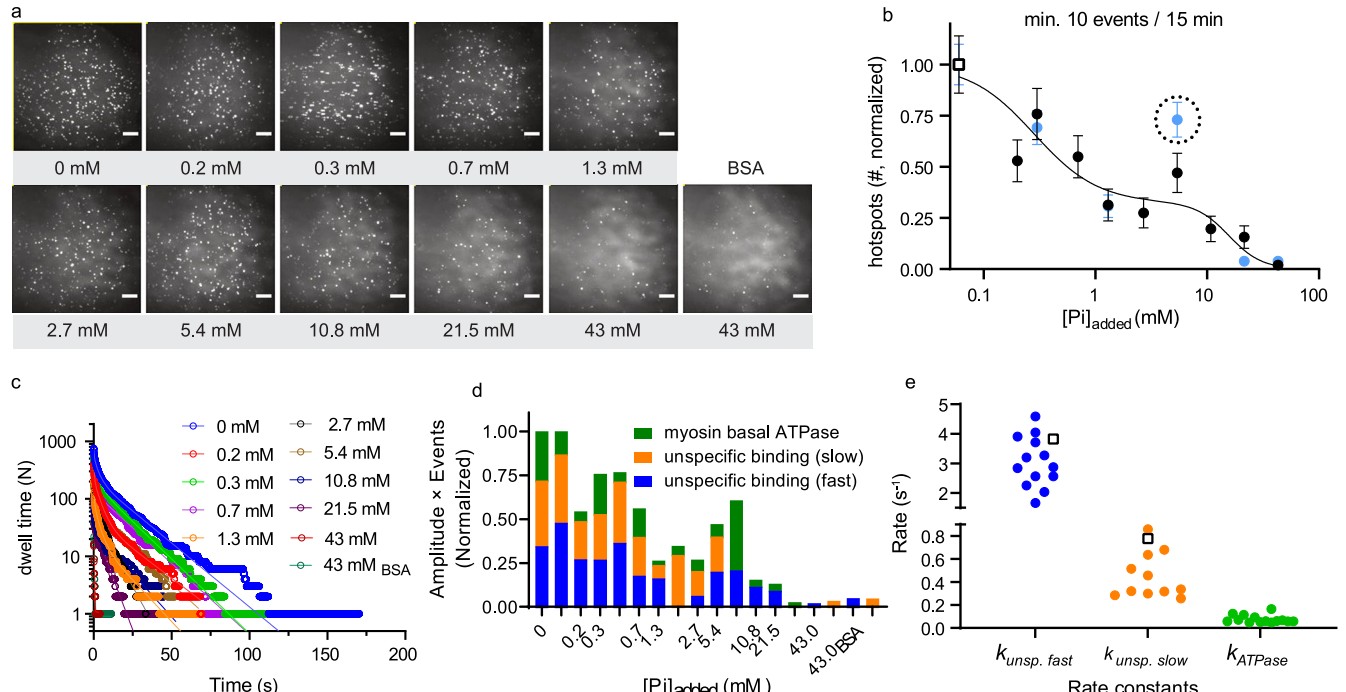

**Fig. 2 | Single-molecule Alexa647-ATP binding to HMM with increasing [Pi].**
**a** Time-averaged Alexa-ATP binding event is represented as fluorescence projections (standard deviation function in Fiji [ImageJ]) of 15 min videos (50 ms exposure time/frame; Movie S1) under increasing added [Pi] (mM). HMM incubation at 34.3 pM. In control experiments, the coverslips were coated only with bovine serum albumin (BSA). Scale bar, 5 μm. **b** Hotspot numbers decrease by increasing added [Pi]. The experiments were performed on two different occasions (series 1, black; series 2, light blue). The numbers (N, point estimates in counting process) were normalized to the number of hotspots (series 1 = 51 and series 2 = 26) in control experiments (open square) without added [Pi], but with estimated background [Pi] ≤0.06 mM. Error bars represent standard deviation (SD) in a Poisson process, estimated as √N. The curved line added to guide the eye represents a fit to the sum of two Hills equations[39] with $K_{d1}$ -0.3 mM and $K_{d2}$ -16 mM. Experiment with HMM

and control experiment (BSA) at 43 mM added [Pi] yielded the same number of hotspots (1). Outlier indicated by dashed circle was not included in the fitting. **c** Cumulative frequency distribution of Alexa-nucleotide dwell time events (series 1) on HMM surface hotspots under increasing added [Pi] were best fitted (solid lines) with triple exponential (0–5.4 mM), double exponential (10.4 and 22.5 mM) or single exponential functions (43 mM). **d** Fractional phases from fittings in **c** normalized to the number of events in control distribution without added Pi ($N_{dwell,series1}$ = 745, $N_{dwell,series2}$ = 374). Note that the amplitudes of all phases were reduced by increasing [Pi]. Slow unspecific ATP binding phase (orange) is not well resolved when [Pi] > 5.4 mM and myosin basal ATPase phase (green) is not detected at 43 mM [Pi]. Best fit mean values ± 95% CI in Supplementary Table 1. **e** Rate constants obtained from fittings to data in **c**. Empty squares are from the fitting of the backgrounds (BSA). Temperature: 23 °C.

actomyosin states e.g., if the charge distribution around the back door exit is modified by actin binding and/or formation of the super relaxed state of myosin[40]. Further, $K_C''$ may change with altered pH, e.g., in muscle fatigue where [Pi] is increased and pH reduced[17,41]. Finally, as already illustrated in Fig. 1, the back door does not exist in all actomyosin states (e.g., in the pre-powerstroke state) and Pi-binding to a secondary site does not exist in the rigor state of myosin VI[16].

## A new contraction model and its key predictions

As suggested previously, the release of Pi from the myosin active site before the powerstroke predicts a reduced isometric force and a minimally changed maximum shortening velocity upon increased [Pi][19,28] but not the experimentally observed [Pi]-independence of mechanical transients[10,27]. Conversely, Pi-release after the powerstroke predicts the [Pi]-independence of the mechanical transients but not the experimentally observed effects of [Pi] on force and velocity[28]. These findings, as well as the Pi-appearance in solution after the power-stroke[7,8], must all be explained by a credible mechanism relating Pi-release to the power-stroke.

A comprehensive and rigorous evaluation of the functional implications of different mechanisms is possible using multi scale mechanokinetic models of muscle contraction. Such models relate molecular properties (strain-dependent actomyosin interaction kinetics, elastic properties, and coarse grain structure) to the contractile function of large actin-myosin ensembles[42–47]. Thus, they offer unique opportunities to quantitatively evaluate contractile

consequences (mechanical transients, ensemble force, velocity, etc.) of different molecular mechanisms of Pi-release. Using models of this type, of different complexity (Supplementary Figs. 5, 6 and Supplementary Table 3–7), we first confirmed the picture laid out in recent studies, that [Pi]-independence of both mechanical transients and maximum shortening velocity is incompatible with each other[10,19,27,28]. Thus, whereas the models in Supplementary Figs. 5a, c, e, g and 6a, c, e, g, assuming Pi-release before the power-stroke, predict appreciable monotonous reductions in isometric force with increased [Pi] and minimal [Pi]-dependence of shortening velocity (Supplementary Figs. 5e, 6e) they do not predict [Pi]-independence of the power-stroke rate (Supplementary Figs. 5g, 6g). The opposite applies to models assuming Pi-release after the power-stroke (Supplementary Figs. 5b, d, f, h and 6b, d, f, h).

In the models in Supplementary Figs. 5, 6, that reflect the two contrasting prevailing views, we assumed a high Pi-release rate (>1000 s⁻¹). Strikingly, the challenges that face models with Pi-release from the active site after the powerstroke would become even more severe if the rate of Pi-release ($k_{p+}$) is as low as 40–120 s⁻¹, suggested by solution kinetics experiments[7,16,32]. Thus, with $k_{p+}$ in the latter range, the model in Supplementary Fig. 6b predicts (Supplementary Fig. 7) both a maximum velocity of shortening at high [MgATP] and a sensitivity of velocity to reduced [MgATP] that are orders of magnitude lower than found experimentally. Despite these severe issues, models of the types in Supplementary Figs. 5b and 6b, with slow Pi-release after the powerstroke, have dominated the literature in recent

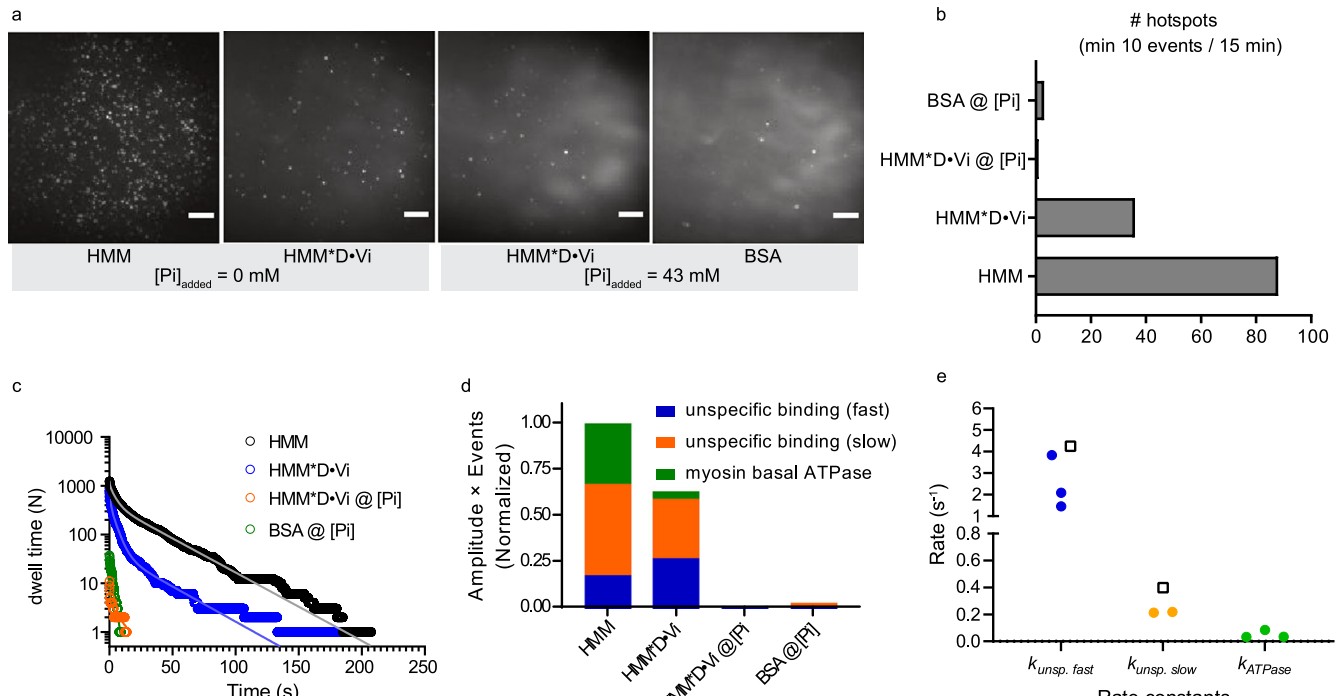

**Fig. 3 | Pi-mediated competitive inhibition of Alexa647-ATP binding to HMM with nonfluorescent ADP (D) locked to the active site by the presence of vanadate (V$_i$). a** Time-averaged Alexa647-ATP binding reflected in fluorescence projections (Zprojection standard deviation in ImageJ) of 15 min videos (50 ms exposure time/frame) under 0 and 43 mM added [Pi] after preincubation with HMM (34.3 pM) with or without pretreatment with ATP and vanadate (HMM:ATP:vanadate; 1:117:583 molar ratios). In control experiments, the coverslips were coated only with bovine serum albumin (BSA). Note a decrease in the total number of Alexa647-ATP binding spots per image with added Pi. The observed decrease in spot number between HMM and HMM*D·Vi (at 0 mM added Pi) should be treated with caution since some of the heads could be lost during HMM*D·Vi complexes preparation for different reasons[35]. Scale bars, 5 μm. **b** Hotspots number estimation under different experimental conditions. **c** Cumulative frequency distributions of Alexa-nucleotide dwell time events on HMM surface hotspots. Data either with or without nucleotide pocket blocked by nonfluorescent ADP (D) and vanadate (HMM*D·Vi complex). The data were best fitted (solid lines) by triple exponential functions (no added Pi) or double-exponential functions (at 43 mM added Pi). **d** Fractional phases from fittings in **c**, under denoted experimental conditions, normalized to the control distribution (HMM, without added Pi, $N_{dwell}$ = 1233). Note that blocking of nucleotide pocket by ADP·vanadate only reduced myosin basal ATPase phase (green), as expected. In contrast, the addition of Pi also reduced amplitudes of unspecific Alexa647-ATP binding phases (blue, orange). Please see Supplementary Table 2 for best fit mean values ± 95% CI. **e** Rate constants obtained from fittings to data in **c**. Empty squares are from the fitting of the background (BSA). Temperature: 23 °C.

years[7,9,10,12,25,27], possibly because a limited set of experimental results, not including velocity, have been considered (however, see ref. 27).

The binding of Pi outside the myosin active site[16] (Figs. 1–3) may explain how Pi appears in the solution after completion of the powerstroke even if Pi leaves the active site before the stroke. Remarkably, we find that incorporation of this idea into a multi scale mechanokinetic model for striated muscle myosin II operation (Fig. 4a, b) allows us to account for the contractile phenomena previously seen as contradictory (e.g., Supplementary Figs. 5 and 6). Thus, first, the model predicts the experimentally observed relationships between [Pi], on the one hand, and isometric force and velocity on the other (Fig. 4c). Second, the model predicts the [Pi]-independence of the powerstroke rate as well as the negligible, [Pi]-independent, delay between crossbridge attachment and the powerstroke (Fig. 4d)[10,27]. In the model in Fig. 4, we only include the secondary Pi-binding site and not any additional sites. The latter may play role in the regulation of Pi-affinity (see above) and modulate the kinetics of the Pi-release (cf. Fig. 5 and Supplementary Fig. 8).

The lack of [Pi]-effects on both velocity and the power-stroke rate in this model (Fig. 4c, d), as well as the combination of a high velocity and slow appearance of Pi in solution after the power-stroke, reflect the central idea that Pi-binding to the secondary site does not affect rates of inter-state transitions along the second and third rows in the scheme in Fig. 4a. I.e., these transitions (e.g., the power-stroke and ADP release) occur at the same rates whether Pi is bound to the secondary site or not.

Importantly, despite a rather slow Pi-exchange governed by the rate constant $k_{p+}$, the model in Fig. 4 also accounts for the isometric tension changes in response to step changes in [Pi] (Pi-transients)[9,13,15] (Fig. 5). Finally, it faithfully reproduces (Supplementary Fig. 9a–d) the shape of the experimental force-velocity relationship. It also reproduces (Supplementary Fig. 9e) the relationship between myosin propelled actin gliding velocity and [MgATP] as faithfully as previously shown for models similar to those in Supplementary Figs. 5, 6[19,20]. See Supplementary Figs. 10, 11 and legends of Supplementary Figs. 9–11 for more details.

Our multi scale mechanokinetic model predicts that the power-stroke can occur with just ADP at the active site (lower row in Fig. 4a). We used high-speed AFM (hs-AFM; Fig. 6; Supplementary Figs. 12–16, 18, 19, and Supplementary Movies 2–17) to test this critical prediction, applying para-aminoblebbistatin (PAB; cf. refs. 20,48,49) for increased longevity of the pre-powerstroke state such that it can be readily captured in hs-AFM scans. The blebbistatin group of compounds, unlike vanadate, bind to myosin outside the active site without direct competition with product or substrate binding. In the absence of PAB, we observed that structural states of the myosin head with ADP at the active site were dominated by an angle of the lever arm close to that seen under ADP conditions (Fig. 6a, b and Supplementary Fig. 12). However, the angle switched transiently to the pre-power-stroke configuration (Fig. 6b–d). The pre-powerstroke configuration became more heavily populated in the presence of vanadate, consistent with the ADP-vanadate state as an analog of the ADP-Pi-state (Fig. 6 and

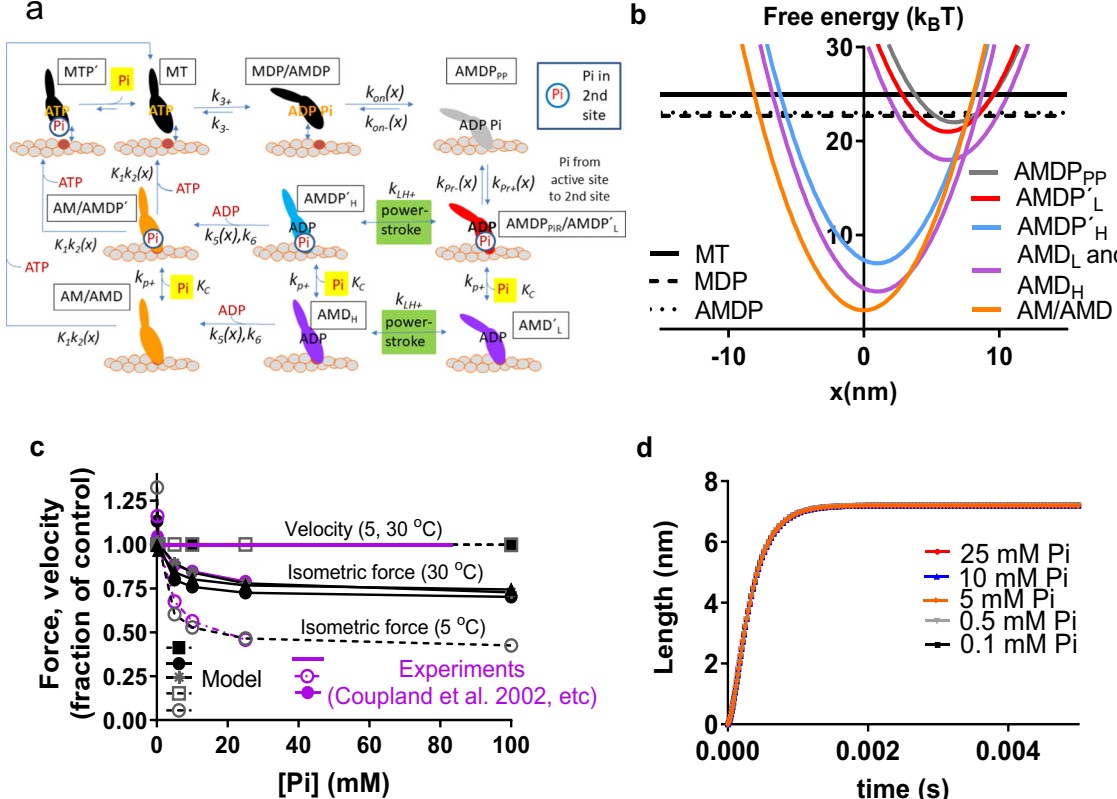

**Fig. 4 | A mechanokinetic model incorporating a secondary Pi-binding site outside the active site. a** A kinetic scheme with Pi-binding outside the active site, as indicated by circles in the second row. The Pi-dissociation rate constant ($k_{P+}$) from the AMDP´$_L$, AMDP´$_H$ and AM/AMDP´ states (blue, red and orange in the middle row) to the AMD$_L$, AMD$_H$ and AM/AMD states and the reverse second order association rate constant $k_{P-}$´ are related as $K_C = k_{P+}/k_{P-}$´. Thus, a pseudo-first-order Pi-association rate constant, $k_{P-} = k_{P+}[Pi]/K_C$ applies for a given Pi-concentration. Short notations for states, as also used in Fig. 4b and elsewhere, are indicated in boxes. **b** The free energy vs the cross-bridge strain variable $x$ for critical states of the model at 0.5 mM Pi. The Pi in the AMDP´$_L$, the AMDP´$_H$, and the (AM/AMD)P´ states

is assumed to be bound to sites outside the active site. The Pi in these states is released at a similar rate ($k_{P+}$) from each of the states. Black straight lines refer to detached cross-bridge states. **c** Simulated steady-state values for the maximum isometric force (black circles) and maximum velocity of shortening (black squares) compared to experimental data (purple) for the velocity[11,104] (10 °C) and force[5] at 30 and 5 °C. The steady-state isometric force for the simplified model in Supplementary Fig. 8 is also shown (gray stars). **d** Simulation of power strokes (nm displacement vs time) for an ensemble of myosin heads attaching in AMDP$_{PP}$ state and cross-bridge force clamped to 0 pN. For further details of the model, see Methods and Supplementary Tables 5–7.

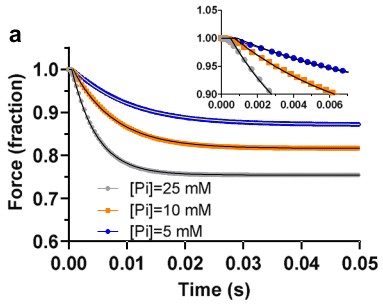

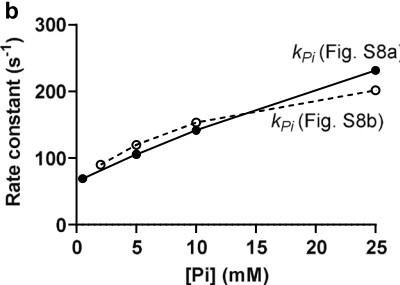

**Fig. 5 | Modeling of key features of the isometric tension changes in response to sudden changes in the Pi-concentration (Pi-transients). a** Simulated tension time course (normalized) for the model in Supplementary Fig. 8a upon a sudden increase in [Pi] from 0.1 mM to the concentration indicated. Curved black lines represent fits to the data of a single exponential function with a time lag ($\Delta t$). Inset: the early phase of tension change and time lag illustrated on an expanded time base. **b** Model simulations of the rate constant of tension change ($k_{Pi}$) after a sudden jump in Pi-concentration vs the final [Pi] for the models in Supplementary Fig. 8a, b. These are simplified versions of the model in Fig. 4 (useful for isometric contraction; cf. gray stars and triangles in Fig. 4c) with the benefit that simulations of Pi-transients are

possible using ordinary differential equations instead of partial differential equations. Note that, due to variability in available experimental data[6,9,10,13,15], no experimental results are included. Both models (Supplementary Fig. 8a, b) predict detailed features of the Pi-transients: 1. single exponential time courses[9] (a), 2. brief delay (<1 ms) for the effects of altered [Pi] on tension[8] (a, inset), and 3. increased rate ($k_{Pi}$) with increased [Pi] in the range 0.1–25 mM (b). The predictions of the model in Supplementary Fig. 8a that $k_{Pi}$ saturates only at very high [Pi] (>10 $K_C$; full line in b) is consistent with results in ref. 13 and ref. 15. The model in Supplementary Fig. 8b, assuming two binding sites outside the active site, is instead consistent with data[9], showing signs of saturation of $k_{Pi}$ at [Pi] <10 mM (dashed line in **b**).

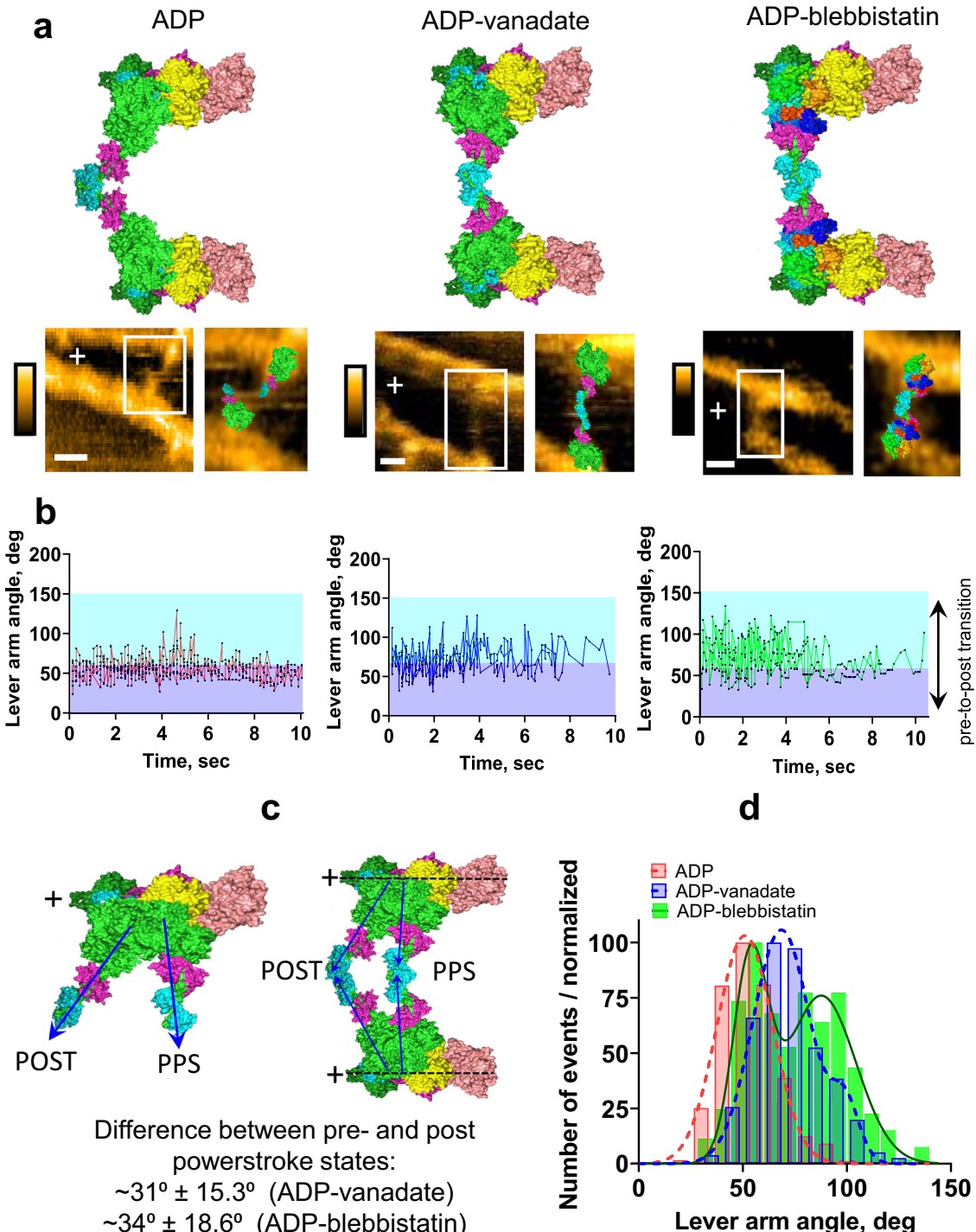

**Fig. 6 | Myosin (HMM) molecules simultaneously attached to two different actin filaments in the presence of ADP and either metavanadate or para-aminoblebbistatin. a** Atomic models of myosin S1-ADP (PDB 3I5F), S1-ADP-VO$_4$ (PDB 1QVI), and S1-ADP-Blebbistatin (PDB 6Z7U with lever arm in pre-powerstroke configuration) and hs-AFM images of myosin heads simultaneously bound to two different actin filaments in the presence of ADP, ADP-metavanadate and ADP-para-aminoblebbistatin (PAB); scale bars: 30 nm. **b** Lever arm angles plotted as a dynamic change of lever arm configuration collected for 10 s. Time resolution: 150 ms/frame (6.7 fps). Red: F-actin-HMM-ADP (number of events: 439). Blue: F-actin-HMM-ADP-metavanadate (number of events: 272). Green: F-actin-HMM-ADP-PAB (number of events: 266). The border between purple and blue areas indicates the mean angle for the lever arm in the ADP configuration obtained from Gaussian distributions. **c** Molecular models of the pre-powerstoke (PPS, presence of ADP-metavanadate or

ADP-PAB) and post-powerstroke (POST, presence of ADP) states with the experimentally measured difference in lever arm configuration. **d** Data from three independent experiments for conditions with ADP (1–10 μM), number of events: 848 ADP (10 μM) + metavanadate (100 μM), number of events: 642 or ADP (10 μM) + PAB (10 μM), number of events: 339 pooled together, showing distinctly different distributions of lever arm angle between ADP: 51.0° ± 13.42° (SD), $r^2 = 0.99$; ADP-metavanadate: 68.6° ± 13.8° (first peak), 99.2° ± 6.7° (second peak), $r^2 = 0.99$; and ADP-PAB: 53.6° ± 8.3° (first peak), 87.7° ± 16.6° (second peak), $r^2 = 0.98$. Time resolution: 150 ms/frame (6.7 fps). The averaged difference in the lever arm position between the strongly bound (post-powerstroke) and weakly bound (pre-powerstroke) states for ADP-metavanadate was: 99.2 ± 6.7°−68.6 ± 13.8° = 30.6 ± 15.3° and for ADP-para-aminoblebbistatin: 87.7 ± 16.6° − 53.6 ± 8.3° = 34.1 ± 18.6°, respectively. Data from Supplementary Movies 2, 4, 5, 7, 11, 15, 17.

Supplementary Fig. 13). However, also in this case, the myosin lever arm dynamically switched between pre-and post-powerstroke configurations. When interpreted vis-à-vis the model in Fig. 4a, the result is consistent with reversible transitions of vanadate ("V") between the active site and the secondary site downstream, with the AMDV$_{PP}$ state, corresponding to the AMDP$_{PP}$ state in Fig. 4a (cf. ref. 16 on similar predictions for Pi). When both 10 μM PAB and 10 μM MgADP were added to actomyosin, the distribution of states with different lever arm configurations became similar to that in the presence of vanadate (Fig. 6d; see also Supplementary Figs. 13, 14). Furthermore, strikingly, as a sign of repeated back and forth transitions between the pre- and post-powerstroke state (corresponding to the lower row in Fig. 4a), the lever arm of a given myosin motor frequently fluctuated between the two configurations (Fig. 6b). Whereas there are some uncertainties in the interpretation of the movies, it is important to note that several individual transitions between pre- and post-power-stroke states in the presence of vanadate or PAB were largely free of ambiguities. One example is given in Supplementary Fig. 15. Other examples can be found by examining Supplementary Movies 4–7 (particularly Supplementary Movies 5–7) frame by frame.

The challenge associated with clear visualization of the lever arm, and measuring its dynamic position, is one of the reasons we decided to use the experimental approach with two HMM heads simultaneously bound between two parallel actin filaments. This approach helped us to achieve higher spatial resolution with a visible lever arm in each myosin head, as previously found in the presence of ATP[50]. Another reason for using this approach is that it allows us to study the lever arm movement in a given molecule through successive hs-AFM frames with greater certainty. When HMM heads are bound to one actin filament in the pre-powerstroke state there is a risk that they can detach and change position on the actin filament (although less likely in the absence of ATP). That is, a given binding site could be occupied by different myosin molecules, especially under weak binding conditions (presence of ADP-vanadate or ADP-blebbistatin). Thus, observation of the myosin head binding to only one actin filament potentially introduces greater uncertainties in the analysis of the lever arm movement. Nevertheless, a comparison of the two experimental approaches (Supplementary Fig. 16) suggests no major differences in key results. Also, a lack of effect of different scanning speeds on measured myosin lever arm angles was verified (Supplementary Fig. 16).

## Discussion

Our results corroborate the idea[16] that Pi-release from the active site precedes and triggers the power-stroke but that Pi does not appear in the solution until after a delay due to binding to the secondary site(s) in the Pi-release tunnel (Fig. 1c). Limitations in the evidence for this view are discussed below. The results also corroborate our hypothesis that there are additional Pi-binding sites on the myosin surface, which may contribute to the delayed appearance of Pi in solution after leaving the active site. The data are consistent with the general picture laid out in[16] but our evidence for more than one Pi-binding site outside the active site suggests higher complexity with versatile control of the Pi-release process. Moreover, our quantitative analysis (Fig. 4) leads to additional testable predictions, which we corroborate by comparison to both new and previously available experimental data. First, the model leads to a remarkably faithful prediction of force- and motion properties of actomyosin (Figs. 4, 5 and S9). Most strikingly, this includes the Pi-independence of both shortening velocity and power-stroke rate, findings that have previously been challenging to reconcile. Second, the described model is supported by our hs-AFM data (Fig. 6) and, less directly by previous optical tweezers experiments using myosin II[30] and myosin V[51]. These studies all suggest that the powerstroke can occur in a biochemical state without Pi, i.e., with ADP only at the active site, just as predicted by our model. Our model also accounts for

previous findings of a significant population of actomyosin states with both Pi and ADP at the active site (e.g., the AMDP´$_H$ and AMDP´$_L$ states in Fig. 4a) in myofibrillar ATP turnover[52,53] and a delayed Pi-release compared to tension development[53]. These findings were previously interpreted differently, i.e., to suggest that Pi-release follows the power-stroke. Indeed, the capability of our mechanokinetic model to account for a wide range of experimental data (Figs. 4,5) that previously appeared contradictory is independent evidence to support the role of Pi-binding sites outside the active site and a multistep Pi-release process because the latter assumptions are critical to the success of the model. A limitation of the model is that it is not fully defined quantitatively. Thus, it would be desirable to devise a method to more exactly measure the Pi-off rate constants from all Pi-binding sites outside the active site.

Manipulation of the amino acid residues involved in the relevant secondary and tertiary Pi-binding sites may also be informative, although possibly complex to interpret. For instance, the R243 residue in the cardiac β-myosin has an important role in the secondary Pi-binding (Fig. 1c). However, it also has a critical role in myosin catalysis and motor function by forming a salt-bridge with E466 in switch II. This salt-bridge closes the back door in the pre-power-stroke state and other preceding states following nucleotide binding and is essential for an intact active site and ATP hydrolysis[54,55]. It is therefore not surprising that mutations of the R243 residue (e.g., R243H and R243C in cardiac β-myosin) and corresponding residues of other myosin isoforms lead to pathologies[56]. However, due to the important roles of the salt-bridge in ATP hydrolysis, it is not possible to use site-directed mutagenesis applied to R243 alone for studying the importance of the secondary Pi-binding site. Interestingly, however, the salt-bridge and effective ATP hydrolysis would be expected to be largely spared by the double mutations E466R and R243E as suggested by studies of corresponding mutations in Dictyostelium myosin II[55]. This result suggests a way to test the importance of the secondary Pi-binding site because the R243E mutation, in the case of double mutants, would be expected to disrupt the secondary site with minor effects on ATP hydrolysis.

For further testing of the generality of the proposed mechanism, it would be of interest to apply the structural studies previously applied to myosin VI[16] to myosin II in order to experimentally verify the binding of Pi to a secondary site in the same myosin class where the biophysical results have been obtained. Conversely, the biophysical studies of the powerstroke and Pi-release using myosin II and V[7,8] would be of interest to apply to myosin VI.

One may ask why secondary Pi-binding should be sufficiently important to be evolutionarily conserved over the myosin super-family. The partitioning of the free-energy change associated with Pi-release and rebinding and the associated versatility in controlling this process could be one factor. Another consideration is that re-formation of the E466-R243 salt-bridge would be expected to be inhibited as long as Pi is bound to the secondary site. This may be important as switch II movements seem to be coupled to lever arm swing mediated via structural changes of the relay helix[57]. Finally, the secondary Pi-binding would have physiologically important roles by preventing rapid Pi rebinding to the active site after Pi-release. This might otherwise lead to a delayed powerstroke or a reduced steady-state velocity, as illustrated by the simulations in Supplementary Figs. 5–7. Such an effect would be of particular importance after the accumulation of Pi in the cytosol, a key factor in exercise-induced muscle fatigue. From a medical perspective, the insights into the Pi-release mechanism is important considering disease-causing mutations in a key residue (e.g., R243 in MYH7) involved in the Pi-release process (Fig. 1c). Furthermore, such insight is also important because the Pi-release is targeted by several recently developed small molecular myosin modulators[49,58–60] with therapeutic potential e.g., in cancer[61], heart failure[62], cardiomyopathies[63], and skeletal muscle disorders[64]. However, most importantly, myosin is not only a critically

important molecular motor in its own right, for which the mechanism coupling force generation and Pi-release has been sought for decades[4–28]. It is also a general model system for chemomechanical energy transduction in biology[7,16]. Insight into the key mechanistic features that fine-tune the operation of such an enzyme is of broad general value. Indeed, the transfer of product from the active site to the bulk solution via several intermediate sites could represent a general mechanism for tuning enzyme operation to specific needs. Interestingly, similar phenomena have been suggested for channeling substrate to the active site of both myosin[35] and other enzymes[39,65].

## Methods

### Ethical statements

The use of animal material for myosin preparation in single-molecule fluorescence studies was approved by the Regional Ethical Committee for Animal experiments in Linköping, Sweden, reference number 73–14. The ethical protocol for use of animal material for the hs-AFM experiments was approved by McGill University, reference number MCGL-5227.

### Chemicals

Alexa Fluor647-ATP (cat. no. A22362) was obtained from Thermo Fisher Scientific. Trolox (cat. no. 238813), cyclooctatetraene (COT, cat. no. 138924), 4-Nitrobenzyl alcohol (NBA, cat. no. N12821), pyranose oxidase (POX, cat. no. P4234), bovine serum albumin (BSA, high purity, cat. no. A0281), dithiothreitol (DTT), catalase (cat. no. C100), creatine phosphate (PK, cat. no. P7936), adenosine triphosphate (ATP, cat. no. A2383), creatine phosphokinase (CPK, cat.no. C3755), MOPS, KCl, MgCl$_2$, K$_2$EGTA, HCl, KOH, Methanol, Glucose, DMSO, Sodium Orthovanadate, Sodium metavanadate, Methylcellulose, Adenosine diphosphate (ADP, cat no. A2754) were of analytical grade and purchased from Sigma-Aldrich (now Merck). The lipids 1,2-dipalmitoyl-*sn*-glycero-3-phosphocholine (DPPC) (cat. no 850355 C), 1,2-dipalmitoyl-3-trimethylammonium-propane (DPTAP, cat. no 890870 C) and 1,2-dipalmitoyl-*sn*-glycero-3-phosphoethanolamine-*N*-(cap biotinyl) (biotin-cap-DPPE, cat. no. 870277) were obtained from Avanti Polar Lipids. Para-aminoblebbistatin (PAB) was obtained from Motorpharma (Hungary). Any other biochemical reagents were of analytical grade and purchased from Sigma-Aldrich (now Merck).

### Single molecule fluorescence measurements of Pi-binding to myosin motor fragments

The single-molecule experiments were performed and analyzed using our established protocol[35] with essential details given below.

**Protein preparations.** Myosin from fast muscle was obtained from New Zealand white rabbits (female, 2 kg, 8–9 weeks). Myosin and HMM were prepared using published protocols (ref. 66, with modifications in ref. 67). Protein preparations were characterized by sodium dodecyl sulfate–polyacrylamide gel electrophoresis, with respect to purity[20], and concentrations, were determined spectrophotometrically.

**Phosphate buffer (PB) for single-molecule fluorescence experiments.** A phosphate buffer was prepared as 1.0 M KH$_2$PO$_4$/K$_2$HPO$_4$ solution, pH = 7.4 at room temperature (22–23 °C). Measurements of the [Pi] concentration using a Phosphate assay Kit (Colorimetric, Cat. #. Ab65622, Abcam, UK) gave good agreement with the intended phosphate buffer stock concentration: [Pi]$_{measured}$ = 1.07 M.

**Determination of background phosphate solution contamination levels for single-molecule fluorescence experiments.** Plastic labware was used throughout and glassware was not permitted. Using the Phosphate assay kit we have determined the background Pi levels in our solutions (Supplementary Fig. 17). Our MilliQ water contains low free [Pi] (<2 μM); consequently, the solutions prepared out also contained Pi in a similar low range. Adjustment of pH using a standard pH meter is considered another source of contaminating Pi. Before pH measurements, the electrode was therefore washed with excess MilliQ water. The LISS ("low ionic strength solution") buffer (10 mM 3-(*N*-morpholino)propanesulfonic acid (MOPS), 1 mM magnesium chloride (MgCl$_2$), and 0.1 mM potassium ethylene glycol-bis(β-aminoethyl ether)- *N*,*N*,*N′*,*N′*-tetraacetic acid (K$_2$EGTA)) after pH adjustment, contained similar lever of Pi as MilliQ water (<2 μM). We also determined the Pi level in the full assay solution yielding ~1.6 mM. After screening all the assay solution ingredients, we found that the contaminating Pi originated in the creatine phosphate stock (CP, part of the ATP regenerating system). The other explanation could be incompatibility (crosstalk) between the phosphate assay kit used and CP. Therefore, we reduced [CP] from standard 2.5 mM to 25 μM final concentration. We reasoned that the ATP regenerating system would still be efficient as the Alexa647-ATP concentration was kept constant at 5 nM. Such adjusted assay solution had [Pi] background level equal to or below 60 μM, considered low enough for our experiments.

**Assay solution for single-molecule fluorescence experiments.** As an assay solution, we have used our optimized TIRF microscopy (TIRFM) buffer[35]. First, a Trolox (Trolox/Trolox-Quinone; TX/TQ) mixture was prepared by dissolving Trolox in LISS and exposing it to UV-light (254 nm) for 15 min to form Trolox-Quinone[68] in order to achieve final assay concentration of TX/TQ ~2 mM. The solution was further supplemented with 10 mM DTT, 7.2 mg/ml glucose, 3 U/ml POX, 0.01 mg/ml catalase, 2.5 mM CP, 0.2 mg/ml CPK, 2 mM COT, 2 mM NBA, 0.64% methylcellulose, and 5 nM Alexa647-ATP. The assay solution was modified from the previously used version[35] to increase the ionic strength from 60 to 130 mM by addition of KCL or Pi.

A concentration of 5 mM KCl was always kept in the assay solution, while 110 mM KCl was gradually replaced with phosphate buffer to increase the Pi-concentration. To estimate phosphate buffer ionic strength, we have used web tools available: https://www.liverpool.ac.uk/pfg/Research/Tools/BuffferCalc/Buffer.html, https://www.iue.tuwien.ac.at/phd/windbacher/node63.html.

With the help of these tools, one can extract molar ratio of phosphate buffer to ionic strength as ~1:2.58 (at pH = 7.4 and T = 23°). Thus, the highest phosphate buffer concentration used in the assay solution was 43 mM (final), as this yielded an ionic strength of ~110 mM. Serial dilutions (1:1) were then performed to achieve other Pi testing concentrations to minimize pipetting error.

Due to increased ionic strength and rather long observation time (15 min videos), we have included 0.1 mg/ml BSA in the assay solution in order to keep the surface fully passivated (counteract any BSA dissociation from the surface and thus minimizing Alexa647-ATP surface binding[35]). As described above, the [CP] was reduced from 2.5 mM to 25 μM in order to keep the background Pi low (<60 μM). The fluorescent nucleotide (Alexa647-ATP) was kept constant at 5 nM (final).

**Experimental procedure for single-molecule fluorescence experiments.** The experiments were performed using our established protocol[35]. Glass coverslips (#1 or #1.5, 24 × 60 mm$^2$) were plasma cleaned using Femto Standard (Diener electronic GmbH, Germany), at 100 W (40 kHz), 0.6–0.8 mbar pressure, for 3 min. Subsequently, coverslips were further cleaned in piranha solution (5 min, 80 °C) and then derivatized with trimethylchlorosilane (TMCS)[69,70]. Caution! Piranha solution is a highly corrosive acidic solution, which can react violently with organic materials. Do not store in a closed container, and use appropriate safety precautions. To passivate surfaces, high purity BSA was centrifuged (220,000×*g*, 15 min) before use[71].

The flow cells were assembled with the abovementioned silanized coverslips for the floor and untreated glass coverslips (#0, 18 mm$^2$ × 18 mm$^2$) for the ceiling of the cell. The floor and ceiling coverslips were

spaced (~100 μm) with double-sided tape (3 M Scotch). The addition of solutions to the flow cell was achieved by adding a 20–30 μl drop of solution at one end of the cell, followed by suction using a filter paper at the other end, with care taken to avoid air bubbles. Flow cells were incubated with rabbit-heavy meromyosin (HMM, 34.3 pM; 5 min) or HMM with the ATP pocket blocked with nonfluorescent ATP in the presence of vanadate, giving HMM*ADP·Vi (denoted HMM*D·Vi below, 34.3 pM; 5 min) (cf. ref. [35]) then with BSA (1 mg/ml; 2 min) followed by MgATP (8 μM, 2 min) to block any irreversible ATP bindings to possible inactive heads (this step was omitted in experiments with vanadate). Subsequently, the flow cells were rinsed using wash solution (LISS buffer with 45 mM KCl and 1 mM DTT) and assay solution was added. Videos were then recorded using our TIRF microscope set-up for 15 min at 19.33 s⁻¹ frame rate[35], an EMCCD camera (Andor iXon Ultra 897 EMCCD), and NIS Elements software (Nikon, ver. 4.51, gain parameter = 100). Video image depth at recording was 16-bit which was then converted to raw 8-bit tiff format to reduce computer time and space needed for processing, analyzing, and storage.

**Data analysis for single-molecule fluorescence experiments.** The experiments were analysed essentially as described previously[35]. Briefly, Fiji/ ImageJ (v. 1.53i)[72] was used to produce time projection images (using Fiji function "Image/Stack/Z-project/STD") and videos of background-subtracted images (using Fiji function "Process/Subtract Background"; "Rolling ball radius" set to five). Brightness and contrast were adjusted by using the Fiji function "Image/Adjust/Brightness/ Contrast". These images served to estimate the total number of Alexa647-ATP binding spots (e.g., Fig. 2a) and to manually locate spot regions (3 × 3 pixels) from which time traces (of each individual spot) were later extracted using Matlab (v. 2020a, 2021a; The MathWorks, Inc. Natick, Ma) or the Fiji function: Image/Stacks/Plot Z-axis Profile. Traces were first screened to assess their general quality by visual inspection. Traces with low signal-to-noise ratio, multistep fluorescence increase/decrease, or similar unclear fluorescence events were excluded. Fluorescence events that started and ended with a one-step change in intensity were kept for further analysis. The prescreened set of traces were finally analyzed by manually measuring dwell time durations from each fluorescence binding spot again using Matlab script. In a majority of cases the dwell time collection was done by another person without knowing the experimental conditions behind them (semi-blind approach).

Further data analysis was done only on so-called hotspots, i.e., spots which exhibited a minimum of ten events (dwells) per 15 min observation time. Collected dwell times from those "hotspots" were plotted as cumulative distributions as described before[73]. The data were then fitted using nonlinear regression in GraphPad Prism (v. 8 and 9) and rates and amplitudes, derived in such fits, were represented as mean ±95% CI (Supplementary Tables 1 and 2). Where needed, errors were also normalized and propagated accordingly.

At first, we attempted to fit the data obtained from HMM to double-exponential functions and the data obtained from HMM*D·Vi complexes to triple exponential functions as before[35]. The slow phase obtained from HMM data represents basal myosin ATPase ($k_{ATPase} \sim 0.05\,s^{-1}$), while the fast phase ($0.2$–$0.5\,s^{-1}$) was explained as unspecific ATP binding to the extra site(s) on myosin with possible functional significance[35]. Data from HMM*D·Vi complexes were best fitted by a triple exponential function. In addition to the first two phases explained above (with basal myosin ATPase phase, now significantly reduced in amplitude), an extra fast phase ($~3.5\,s^{-1}$) of unspecific Alexa647-ATP binding emerged (see also ref. [35]).

We noticed in the present study that, in general, this extra fast phase can be resolved also from Alexa647-ATP data in the absence of vanadate. The reason for that could be higher ionic strength used and/ or other modifications of the assay buffer (inclusion of BSA and reduction of CP concentration) compared to previous work[35].

In order to make the choice of a fitting model as unbiased and consistent as possible, we relied on the GraphPad Prim functionality of comparing and selecting the model to be used (i.e., single, double, or triple exponential model) based on the Akaike's Information Criterion (AICc). In addition, if a fit was flagged as ambiguous by the program or if the CI intervals could not be calculated, we chose the simpler model. In addition to the software mentioned above in this section, we also used MS Excel 2016, in some parts of the analysis.

**High-speed atomic force microscopy (hs-AFM)**
The hs-AFM experiments were performed on a tapping-mode system (RIBM, Japan, Toshio Ando's model), using Olympus cantilevers (BL-AC10DS-A2) with the following parameters: spring constant 0.08–0.15 N/m; quality factor in water ~1.4–1.6 and resonance frequency in water 0.6–1.2 MHz. To achieve better spatial resolution, an additional carbon probe tip was fabricated on the end of the original Olympus cantilever by electron-beam deposition at the Kanazawa University (Kanazawa, Japan) during Bio-SPM collaborative work. The fabricated tip was sharpened by a plasma etcher giving a ~4 nm tip apex (this parameter was used for a simulation of the AFM images[74] using protein structures with the specific lever arm configuration). The free oscillation peak-to-peak amplitude (A0) of the cantilever was set to ~2.0 nm and the amplitude set point was adjusted to ~0.9–0.92 A0. Rabbit fast skeletal muscle heavy meromyosin (HMM) and F-actin were purified as described in refs. 75,76. Prior to hs-AFM experiments, HMM and F-actin were tested for their functionality using in vitro motility and MgATPase activity assays as described in ref. 77.

**Protein preparations.** Animal material was derived from New Zealand white rabbits (female, 2.6–2.7 kg, 11–12 weeks). Myosin was purified from the rabbit psoas muscle using standard procedures[75,76] and HMM was prepared[75] by proteolysis of myosin using α-chymotrypsin (Sigma-Aldrich). Actin was purified from acetone powder of rabbit skeletal muscle using standard protocols[75,76].

**Sample preparation for hs-AFM experiments.** The lipid bilayer used for hs-AFM imaging contained DPPC, DPTAP and biotin-cap-DPPE dissolved in chloroform. The DPPC: DPTAP: biotin-capDPPE lipids were mixed in a weight ratio of 89:10:1. The details of preparing lipid vesicles and depositing them on a 1.5 mm mica disk to create the mica-supported lipid bilayer substrate (mica-SLB) have been previously described in refs. 50,77.

After rinsing the mica-SLB with the hs-AFM buffer, containing 25 mM KCl, 4 mM MgCl₂, 0.5 mM EGTA, 25 mM Imidazole-HCl, 2 mM DTT, pH 6.0), 2.8 μl of 7 μM actin filaments diluted in hs-AFM buffer were deposited on the mica-SLB and incubated for 10 min under a wet cap to avoid drying the lipids. After 10 min, non-attached actin filaments were rinsed away by the buffer and 3 μL of 15 nM HMM was added and incubated for 10 min under a wet cap to form the rigor actin-myosin complex. The substrate was rinsed again with the buffer to remove unbound myosin molecules and the sample stage attached to the Z-scanner was immersed in the AFM experimental chamber. The hs-AFM chamber (volume ~100 μl) was filled with the solutions matching the experimental conditions (presence of ADP, ADP-metavanadate or ADP-para-aminoblebbistatin; ADP-PAB).

For the control experiments, the imaging was performed in the presence of different ADP concentrations in the range of 1–10 μM ADP to test the binding of the myosin molecules to the actin filament (or between two actin filaments) and myosin lever arm configuration. Evaluated ADP concentrations (1, 2, and 10 μM) were enough to prime the myosin lever arm in the strong-binding configuration (~50° relatively to actin filament) with quite narrow standard deviations (~10–15°) between conditions (Supplementary Fig. 18). This finding is consistent with earlier results showing high ADP binding at 10 μM

ADP[68]. Therefore, data for all ADP concentrations were combined for the subsequent analysis of the ADP state. However, for experiments with PAB, 10 μM ADP was consistently used. The concentration of PAB (10 μM) was lower than its myosin association constant due to the deleterious effect on the actin- and HMM-binding lipid bilayer at high concentrations of DMSO (used to dissolve PAB). As a result, only a fraction of the myosin heads are expected to have had PAB bound. However, the average change in myosin head configuration in response to PAB was evident under the conditions used.

To image the F-actin-HMM-ADP-metavanadate and F-actin-HMM-ADP-PAB complexes, the following protocol was used. Prior to adding HMM to the actin filaments, the filaments were attached to the mica-SLB substrate. Then, 15 nM HMM was added and incubated for 30 min at 25 °C with 100 μM sodium metavanadate or 10 μM para-aminoblebbistatin in the presence of 1, 2, or 10 μM ADP.

Preparation of vanadate stock solution was according to ref. 78. Briefly, 200 mM of sodium metavanadate was prepared with pH adjusted to 10.0 and with boiling to reduce the polymerization until the color of the solution changed from yellow to white. An aliquot of the stock solution was diluted in the hs-AFM buffer to attain the desired concentration and was then added to the 15 nM HMM solution. The para-aminoblebbistatin was prepared according to the manufacturer´s recommendations. Briefly, DMSO was added to the lyophilized powder to make a 32.6 mM stock solution. The aliquot of the stock solution was diluted in the hs-AFM buffer to attain the desired concentration that was added to 15 nM HMM solution. After 30 min incubation, the HMM-ADP-metavanadate or HMM-ADP-para-aminoblebbistatin were added to the actin filaments attached to the mica-SLB in the appropriate experimental buffer and incubated for 10 min under a wet cap. The imaging was performed in the presence of 10 mM glucose and 1 U/ml of hexokinase (Sigma-Aldrich, USA) to remove the possible presence of ATP in the ADP solutions.

Measurements of the lever arm configuration of HMM-ADP, HMM-ADP-metavanadate, and HMM-ADP-para-aminoblebbistatin bound to actin filaments was performed in a similar way as described recently[50]. That is, each of the two heads of the HMM molecule is bound to two different actin filaments located roughly in parallel on the mica-SLB. The angle of the lever arm was measured relative to the actin filament. First, the polarity of actin filament was defined by the morphology of the bound myosin heads, where heads formed so-called "arrow-head structures" determined in electron microscopy studies[79]. The arrows in these structures represent the myosin heads with their lever arms directed toward the barbed (+) end, as shown in Supplementary Fig. 18. Second, the x and y coordinates of the myosin head and lever arm were defined using Kodec software (v. 4.4.7.39)[80]. The angle tool featured in ImageJ (NIH, USA) was used to calculate the lever arm angle relative to the actin filament, considering the center of mass of the HMM head calculated from x,y coordinates and the orientation of the HMM heads (hs-AFM Source Data file).

**Data analysis and processing of HS-AFM images.** Data collection was performed using IgorPro software (v. 6.3.7.2, Wave Metrics). To remove spike noise in the image and to make the xy-plane flat, the hs-AFM images were processed with low-pass filtering by Kodec software (4.4.7.39). In case of the uncertain orientation of the lever arm, the image was processed by applying the ImageJ gamma filter with the function f(p) = (p/255)γ × 255 to each pixel (p), where 0.1≤ γ ≤ 5.0, and 3D surface plot plugin with isoline filter that joins points of equal height value. The original and processed hs-AFM images are included in Supplementary Movies 2–7, along with highlighted lever arm orientations. The total number of heads and molecules analyzed were for (i) ADP conditions: 58 HMM heads simultaneously attached to two different actin filaments and 34 HMM molecules bound to one actin filament (ii) ADP-metavanadate conditions: 53 HMM heads

simultaneously attached to two different actin filaments and ~14 HMM molecules bound to one actin filament and (iii) ADP-para-aminoblebbistatin conditions: 15 HMM heads simultaneously attached to two different actin filaments. Values are reported as mean ± standard deviation of the mean.

## Molecular modeling

To generate the model of myosin II with the Switch II domain as in the Pi-release state of Myosin VI[16] and the phosphate in the secondary site, we used the PDB structure 5N6A (cardiac muscle β-myosin II in the pre-powerstroke state) as a template. Homology modeling was performed using the SWISS-MODEL server (https://swissmodel.expasy.org/), with the structure of myosin VI motor domain in the Pi-release state (PDB code 4PFO (Pi-release state)) as a template. The homology model was superimposed on the 5N6A (Pre-powerstroke) structure in the UCSF-Chimera program. Thereafter, residues 461 to 473 were taken from the model and all other residues were taken from the original 5N6A (Pre-powerstroke) structure to generate the structures that are shown in Fig. 1 and Supplementary Figs. 1, 2.

To calculate the possible pathway that the Pi molecule could take through the myosin head, the program HOLE (v. 2.2.005; http://www.holeprogram.org/)[81,82] was used. A point near the ATP binding site was picked and several vectors were tested to determine the direction of the exit tunnel. The output was visualized in the visual molecular dynamics program (VMD; v. 1.9.3; http://www.ks.uiuc.edu/Research/vmd/)[83] with the tunnel colors representing in red a tunnel too small for a water molecule (1.7 Å), in green a large enough for a molecule with radius 2.5 Å and in blue a wider channel. The inorganic phosphate should fit ($PO_4^{3-}$ radius: 2.38 Å) comfortably through the green (and blue) parts of the tunnel. Further optimization of the structure was performed to ensure that the position of the inorganic phosphate is reasonable by running 1000 steps of energy minimization using the CHARMM software Version 45b1[84] by applying 1000 steps of ab-initio Newton Raphson minimization. The CHARMM36 force-field was used[85].

Estimation of the Gibbs energy of binding for the phosphate in the secondary site was performed by running a quantum chemical calculation using the quantum mechanical (QM) cluster approach[86]. This approach was successfully employed by us to study the binding of drugs[87] and ions[88] to proteins. In QM-cluster calculations of binding energies to proteins, the ligand (in this case, phosphate ion, $PO_4^{3-}$) is studied together with the residues that interact with it directly by a full QM calculation, whereas the rest of the system is approximated as a continuum. Here, the side chains of residues Ser180, Thr235, Ser241, and Arg243 were used, and also atoms CA, CB, and CG of Arg 237. Including the phosphate, a coordinating water molecule and all hydrogen atoms, the QM system to consider had 62 atoms. The positions of all atoms were optimized by running geometry optimization in NWCHEM (v. 6.8.1)[89] using the def2-svpd basis set[90] and the COSMO model[91,92] to represent the solvent (water). The binding partners (protein and ion) were optimized in the same manner. Energies were calculated using the larger def2-tzvpd basis set[80] and the SMD solvation model[93] after removing the water molecule (geometry optimization was performed with COSMO rather than SMD as the implementation of COSMO in NWCHEM is more efficient). Thermal corrections to the enthalpy and entropy were calculated using the frequency module in NWCHEM, with the def2-svpd basis set. The meta-hybrid, general purpose M06 DFT functional[94] was used in all calculations. Of note, our previous calculations indicated that representing the solvent as water yielded more accurate estimates of the binding energy than using a lower dielectric solvent[87]. Long-range dispersion was treated with DFT-D3[95]. All calculations were run on Beskow, a Cray XC40 system maintained by the Swedish National Infrastructure for Computation (PDC node, in KTH Stockholm).

## Multi scale mechanokinetic models

**General.** We used what we denote multi scale mechanokinetic models to relate molecular properties, including actomyosin interaction kinetics and actomyosin elastic properties, to the contractile behavior of a large ensemble (such as that in muscle) of actomyosin cross-bridges. This type of model was originally developed by Huxley[43] with a full theoretical formalism later developed by Hill[42]. Further details on how to integrate actomyosin biochemistry with elastic and structural properties of the cross-bridges have been considered more recently[44–47].

All states in the models used here are defined in the main Fig. 4, Supplementary Figs. 5, 6, and Supplementary Tables 3–7. The myosin (M) and actomyosin (AM) states have either substrate (ATP; T) or products (ADP, D; inorganic phosphate, P or Pi) at the active site. We assume three states with myosin either detached from actin or non-stereospecifically, weakly attached (MT, MDP, AMDP). The stiffness in the weakly bound AMDP state is set to zero to account for its negligible effects on maximum shortening velocity under physiological conditions[20]. The stereospecifically attached states are of different biochemical and structural types (Fig. 4). I.e., the subscripts "L" and "H" in AMD$_L$/AMDP$_L$ and AMD$_H$/AMDP$_H$ in main Fig. 4 refer to "low" and "high" force, respectively. Here, the low force state corresponds to that before the main force-generating structural transition (the "powerstroke")[96], whereas the high force state corresponds to a state after the powerstroke. Model parameter values (Supplementary Tables 3–7) are primarily from independent solution biochemistry and single-molecule mechanics experiments[47] with minor modifications (Supplementary Tables 3–7). In order to simulate results for low temperature (5 °C), we assume that the parameter values are changed (Supplementary Table 7) as motivated previously[20]. Additionally, we assume a uniform distance ($x$) distribution between the myosin heads and the center of the nearest myosin binding site on actin (cf. refs. 42,43,97) Furthermore, the stiffness ($k_s$ = 2.8 pN/nm) of each of the two heads of each myosin molecule in strongly bound states is assumed linear (Hookean) unless otherwise stated. Below, any free energy differences between states ($\Delta G_w$, $\Delta G_{AMDP-AMDL}$, $\Delta G_{AMDL-AMDH}$, and $\Delta G_{AMDH-AM}$; see further Supplementary Tables 3, 5) are given in units of $k_BT$ ($\approx 4$ pN nm) where $k_B$ is the Boltzmann constant and T is the absolute temperature.

**Model in Supplementary Fig. 5a slightly modified from Månsson (2019)[19] – Pi-release before power-stroke.** The equilibrium constant, $K_w$ for weak binding of the myosin head to actin is given by $K_w = \exp(\Delta G_w)$ for $-2.8$ nm $< x <$ 18.2 nm and 0 elsewhere. The quantity $\Delta G_w$ denotes the free energy difference between the MDP state and the AMDP state. The next transition in the cycle is that from the weakly bound AMDP state to the first stereospecifically strongly bound but transient (T), AMDP$_T$ state. This rate function is given by:

$$k_{on}(x) = k_{on'} \exp[\Delta G_{AMDP-AMDL} - (k_s/2)(x-x_1)^2/(2k_BT)] \quad (1)$$

for $-2.8$ nm $< x <$ 18.2 nm and 0 otherwise.
with the reverse transition given by:

$$k_{on-rev}(x) = k_{on'} \exp[(k_s/2)(x-x_1)^2/(2k_BT)] \quad (2)$$

Subsequently, Pi is rapidly and reversibly released from the AMDP$_T$ state in a strain-insensitive transition to form the AMD$_L$ state. If the forward, first-order, rate constant is denoted $k_{p+}$ the backward pseudo-first-order rate constant (at constant [Pi]) is given by:

$$k_{p-} = k_{p+}[Pi]/K_C \quad (3)$$

where $K_C$ is the dissociation constant for Pi-binding to myosin and *[Pi]* is the concentration of inorganic phosphate in solution.

The next transition is the power-stroke as originally defined by Huxley and Simmons[96]. The forward (7) and reverse (8) transitions are governed by the rate functions

$$k_{LH+}(x) = k_{LH-}(x) \exp(\Delta G_{AMDL-AMDH} + k_s(x-x_1)^2/(2k_BT) - k_s(x-x_2)^2/(2k_BT)) \quad (4)$$

and

$$k_{LH-}(x) = 2000 \text{ s}^{-1} \quad (5)$$

respectively.

We next assume a strain-dependent transition rate constant from the AMD$_H$ to the AMD state[98–100]:

$$k_5(x) = k_5(x_1) \exp(\Delta G_{AMDH-AM} + k_s(x-x_2)^2/(2k_BT) - G_{AM}(x))) \quad (6)$$

where

$$G_{AM}(x) = \int_x^{x_3} F_{AM}(x'-x_3)dx'/k_BT \quad (7)$$

with $F_{AM}(x'-x_3)$ being a force function with piece-wise constant slope. This transition is here taken as irreversible as motivated by the assumption that [MgADP] $\approx 0$ and the lumping together of the AMD and the AM state to an AM/AMD state.

The detachment rate function from the AMD to the MT state can be approximated by:

$$k_{off}(x) \frac{k_2(x)k_6[MgATP]}{\frac{k_6}{K_1} + (k_2(x)+k_6)[MgATP]} = \frac{k_2(x)[MgATP]}{\frac{1}{K_1} + \frac{k_2(x)}{k_6}[MgATP] + [MgATP]} \quad (8)$$

with

$$k_2(x) = k_2(0) \exp(\frac{|F_{AM}(x-x_3)| \cdot x_{crit}}{k_BT}) \quad (9)$$

where, $k_2(0)$ ($k_2$ in Supplementary Tables 3–7) and $k_6$ are rate constants for ATP-induced detachment from the AMT state at $x = 0$ and ADP-dissociation from the AMD state, respectively. The parameter $K_1$ is an equilibrium constant for MgATP binding to the AM/AMD state (Fig. 4a) and $x_{crit}$ defines strain-sensitivity of $k_2(x)$[92].

**Model in Supplementary Fig. 5b modified from that in Supplementary Fig. 5a to have Pi-release after the power-stroke.** The model in Supplementary Fig. 5b is similar to that in Supplementary Fig. 5a except for the key assumption that Pi-release occurs after instead of before the power-stroke. More specifically, the rate functions $k_{on}(x)$ (Eq. 1) and $k_{on-rev}(x)$ (Eq. 2) govern the transitions between the AMDP and the AMDP$_L$ states in this model because the AMDP$_T$ state is omitted. Furthermore, the new AMDP$_L$ and AMDP$_H$ states in the model in Supplementary Fig. 5b substitute the AMD$_L$ and AMD$_H$ states and are separated by the power-stroke and reverse stroke transitions, governed by $k_{LH+}$ and $k_{LH-}$, respectively. Finally, the strain-independent Pi-release occurs with a rate constant $k_{p+}$ from the AMDP$_H$ state and the reverse rate constant $k_{p-}$ is given by Eq. (3). Apart from the mentioned changes, all rate functions, as defined in Eqs. (4–9) (see also Supplementary Tables 3–4) are similar to those for the models in Supplementary Fig. 5a, b.

**Model in Supplementary Fig. 6a - modified from ref. 20—Pi-release before powerstroke.** This model differs from the model in Supplementary Fig. 5a by introducing the pre-power-stroke state (AMDP$_{PP}$) and a Pi-release state (AMDP$_{PiR}$) from which Pi is rapidly and reversibly released. The notion of these states are due to Llinas et al.[16] and

formalized into a multi scale mechanokinetic model by ref. 20. Further, the model in Supplementary Fig. 6a is simplified by the assumption that the detached state and the weakly and non-stereospecifically bound states are both lumped into one MDP state. The transition between the latter and the AMDP$_{PP}$ state is governed by the rate function:

$$k_{\text{on}}(x) = k_{\text{on}'} \exp(\Delta G_{\text{on}} - k_s(x - x_1)^2/4k_BT) \qquad (10)$$

where $\Delta G_{\text{on}}$ is the difference in free energy minima between the MDP and AMDP$_{PP}$ states.

The reversal of the rate function is given by:

$$k_{\text{on}-}(x) = k_{\text{on}'} \exp(k_s(x - x_1)^2/4k_BT) \qquad (11)$$

The transition into the subsequent Pi-release state (AMDP$_{PiR}$) (19) and its reversal (20) are given by:

$$k_{\text{Pr}+}(x) = k_{\text{Pr}+'} \exp(\Delta G_{\text{PiR}}/2 - (k_s/2)(x - x_1)^2/(2k_BT) \\ + (k_s/2)(x - x_w)^2/(2k_BT))) \qquad (12)$$

and

$$k_{\text{Pr}-}(x) = k_{\text{Pr}+'} \exp(\Delta G_{\text{PiR}})//2 + (k_s/2)(x - x_1)^2/(2k_BT) \\ - (k_s/2)(x - x_w)^2/(2k_BT)) \qquad (13)$$

where $\Delta G_{\text{PiR}}$ is the difference between the free energy minima of the AMDP$_{PP}$ and the AMDP$_{PiR}$ states.

The subsequent, rapidly reversible Pi-release is assumed to be strain-independent and governed by a first-order rate constant $k_{p+}$ with the reverse pseudo-first-order rate constant $k_{p-}$ as defined in Eq. (3) above. The remaining rate functions in the model shown in Supplementary Fig. 6a are defined as the corresponding functions for the model shown in Supplementary Fig. 5a (Eqs. 4–9).

**Model in Supplementary Fig. 6b – Pi-release after powerstroke.** The model in Supplementary Fig. 6b is similar to that in Supplementary Fig. 6a but assumes Pi-release after, instead of before the power-stroke. Furthermore, as a consequence, the rate functions $k_{\text{Pr}+}(x)$ and $k_{\text{Pr}-}(x)$ (Eqs. 12 and 13) instead govern the transitions between the AMDP$_{PP}$ and the AMDP$_L$ states in the model in Supplementary Fig. 6b because the AMDP$_{PiR}$ state of the model in a Supplementary Fig. 6a is omitted. The remaining changes in transforming the model in Supplementary Fig. 6a to that in Supplementary Fig. 6b are identical to those used to transform the model in Supplementary Fig. 5a to that in Supplementary Fig. 5b. Thus, the rate functions for all transitions between the AMDP$_L$ and the MT are given by Eqs. (3–9).

**Model in main Fig. 4 with two-step Pi-release.** The model in main Fig. 4 has appreciable similarities to the model in Supplementary Fig. 6a except for the lumping together of the AMDP$_{PiR}$ state and the AMDP$_L$ states (that are in rapid equilibrium without changes in lever arm angle) as well as introduction of the AMDP'$_L$ and the AMDP'$_H$ states and the associated transitions. In the model in Fig. 4, Pi is rapidly shifted from the active site to secondary sites upon transition from the AMDP$_{PP}$ to the AMDP$_{PiR}$/AMDP'$_L$ state. In the latter state, Pi pauses in secondary site(s) without allosteric effects on the rest of the myosin molecule. The Pi is then rather slowly released to solution with a strain-independent rate $k_{p+}$ leading to an AMD$_L$ (or AMD$_H$) state (see below). The reverse rate constant for Pi-binding is given by $k_{p+}[Pi]/K_C$. As a corollary to the lack of allosteric effects of Pi-binding to the secondary site, the power-stroke and its reversal occur with similar rates between the AMDP'$_L$ and the AMDP'$_H$ states as between the AMD$_L$ and the AMD$_H$ state. Furthermore, the displacements (the power-stroke distances) are assumed to be identical in both cases. Pi is also assumed to be

released from the AMDP'$_H$ and the AM/AMDP´ state with rate constant $k_{p+,}$ with the reverse reaction being governed by $k_{p+}[Pi]/K_C$. Finally, Pi is rapidly and irreversibly released from the MTP´ state.

**Derivation of experimentally observable steady-state variables from models.** Steady-state contraction at velocity, $v$, was simulated by solution of differential equations for the state probabilities (for all $j,k$):

$$\frac{da_j}{dx} = \left(\sum_{k}^{n1} k_{kj}(x)a_k(x) - \sum_{k}^{n2} k_{jk}(x)a_{j(x)}\right)/v \qquad (14)$$

Here, $a_j(x)$ are the state probabilities for the different model states. The rate functions $k_{kj}(x)$ and $k_{jk}(x)$ govern transitions from state $j$ into state $k$ and from state $k$ into state $j$, respectively. The quantity $n1$ is the number of states from which transitions can occur into state j whereas the quantity $n2$ is the number of states to which transitions from state $j$ can occur. The model simulations were implemented by numeric solution of the system of differential equations described by Eq. (14). Observable variables (force and ATP turnover rate) were then calculated from the appropriate state probabilities[101] by averaging over the inter-site distance (36 nm) along the actin filament. Thus, average force $<F>$ (in pN) per myosin head (whether attached to actin or not) is given by:

$$<F> = \frac{\sum_1^{n3} \int_{-22}^{14} k_s(x)a_j(x)(x - x_j)dx}{36} \qquad (15)$$

Here, n3 is the number of actin-attached cross-bridge states. In analogy, the stiffness ($<S>$; Eq. 16) and the fraction of attached myosin heads ($<Na>$; Eq. 17) are obtained as follows:

$$<S> = \frac{\sum_1^{n3} \int_{-22}^{14} k_s(x)a_j(x)dx}{36} \qquad (16)$$

$$<Na> = \frac{\sum_1^{n3} \int_{-22}^{14} a_j(x)dx}{36} \qquad (17)$$

The ATP turnover rate $(<ATPase>)$ is given by:

$$<ATPase> = \int_{-22}^{14} k_{\text{off}} a_3(x)dx/36 \qquad (18)$$

where $a_3$ represents the fraction of all myosin heads that are in the AMD$_H$ and AMDP´$_H$ states. Numerical integration of Eqs. 15–18 starts at $x = 14$ nm and progresses in the negative $x$-direction. At $x = 14$ nm, initial values for all attached states are set to zero and equilibrium distribution is assumed for the MT and MDP states. For stable numerical computations, the values of the rate functions were limited to a maximum ($r_{\max}$) of 100,000 s$^{-1}$ for isometric contraction and $1 \times 10^6$ s$^{-1}$ – $1 \times 10^8$ s$^{-1}$ for the fastest velocities of shortening, and a minimum ($r_{min}$) of $1 \times 10^{-6}$ s$^{-1}$. If outside these limits, the parameter value was set to either $r_{\max}$ or $r_{min}$.

Differential equations were solved numerically using a Runge–Kutta–Fehlberg (4/5) integration algorithm in the computer program Simnon (Department of Automatic Control, Lund Institute of Technology, Sweden) as described in detail recently[97].

**Simulations of power-stroke.** The power-stroke was simulated (e.g., Fig. 4d) based on the assumption that tension is clamped to zero during the entire simulation, thus differing from single-molecule experiments that were performed while clamping the force primarily at positive (>1.5 pN) levels[10]. The basis for using this approach is several-fold: 1. It is simplest to assume zero force, 2. It seems likely that the average strain (and thereby force) is close to zero for a cross-bridge attaching into its first pre-power-stroke state in muscle fiber and 3. The

fast phase of the mechanical transient (the one investigated in the single-molecule studies[10]) has been found to be independent of [Pi] in muscle fibers regardless of whether releases or stretches are studied[6].

In the simulations of the power-stroke, the rate functions above (1–13) were used. Furthermore, consistent with the averaging approach used in the single-molecule study[10] and with muscle fiber experiments[6], the displacement traces were achieved by solving ordinary differential equations in the state probabilities. The detailed approach is described in the following for the model in Supplementary Fig. 5a based on Eqs. 1–9. The principle is similar to that used for the other models tested with different details associated with the underlying kinetics schemes. In all cases, the simulation starts with myosin cross-bridges populating an early pre-powerstroke state expected to be detected in mechanical measurements. In the case of the model in Supplementary Fig. 5a, the state probability for the $AMDP_T$ state is thus set to 1, whereas other initial values for state probabilities are set to zero. By the assumed clamp of tension to zero, the cross-bridge in the $AMDP_T$ state is held at a strain value $x = x_1$ where the free energy of the state is at its minimum. Because we are only interested in evaluating the power-stroke we ignore all detachment events (by reversal of the attachment into the $AMDP_T$ state) in the simulations. That is, we set $k_{on-rev}(x_1) = 0$ (Eq. 2). Cross-bridges in this ($AMDP_T$) state that progresses to the power-stroke will first be shifted to the $AMD_L$ state governed by Eq. (3). Next, the power-stroke occurs with rates governed by Eq. (4, 5). This is associated with a shift of the cross-bridges along the x-coordinate from $x = x_1$ to $x_2$ under the condition that tension is clamped to zero. The next steps are MgADP dissociation and MgATP-rebinding. The latter process was slowed in the simulations (similar to the approach used in single-molecule experiments[10]), by setting [MgATP] = 10 nM. Additionally, reattachment of cross-bridges from the MT to the $MADP_T$ state was not allowed, limiting the simulation to one forward cycle from the state $AMDP_T$ to the detached states. In general, independent of the specific model, the values of the rate functions for transitions from one state into a neighboring state are calculated by inserting a value of $x$ in each rate function for which force is zero in the corresponding state.

Finally, the displacement time ($t$) course (progression of the power-stroke) of the average cross-bridge strain, $\Delta L(t)$ for an ensemble of myosin heads initially attaching in the $AMDP_T$ state at $x = x_1$ (i.e., with force clamped to zero) is given by:

$$\Delta L(t) = ([AMD_L](t)^*(x_1 - x_2) + [AM/AMD](t)^*(x_1 - x_3))/([MDP](t) + [AMDP_T](t) + [AMD_L](t) + [AM/AMD])$$

(19)

Here the ordinary parentheses around $t$, indicate a functional dependence on $t$ and the hook-parentheses indicate probability of the respective state.

**Simulations of isometric contraction and Pi-transients using a simple kinetic scheme.** The Pi-transients were simulated using the simplified kinetic scheme in Supplementary Fig. 8 for isometric contraction, assuming one given average cross-bridge strain, corresponding to $x = x_{II}$ for the model in Fig. 4. Rate constants for all transitions, given in the legend of Supplementary Fig. 8, were approximated from the rate functions for the complete model in Fig. 4 assuming $x = x_{II}$. In the simulations, the initial values for the probabilities of all cross-bridge states were set to 0 except for the probability of the MATP state that was 1. The initial value problem for the system of differential equations was then solved to derive the time course for approach of steady-state isometric force at a given concentration of Pi. In these simulations, force was assumed to be given by

$$\text{Force} = k_s([AMD_H] + [AMDP'_H])$$

(20)

where $k_s$ is cross-bridge stiffness and the hook-parentheses have a similar meaning as in Eq. (19).

The steady-state force value eventually reached in these simulations is also well approximated by an analytical steady-state solution of the differential equations that governs the model in Supplementary Fig. 8 (see inset of Supplementary Fig. 8a), if the detachment at the end of the power-stroke ($k_{det}$) is neglected. The state probabilities at the isometric steady-state were then taken as initial values for simulating the Pi-transients. Re-running of the simulations with the new Pi-concentration and the new initial values now gave the time course of the Pi-transient.

**Sensitivity analysis.** The sensitivity ($S_{ij}$) of each of a number of key output variables ($y_j$) to change in a specific parameter value ($\Delta_{pi}$) is quantified as $\Delta y_j/\Delta_{pi}$ where $\Delta y_j$ is the change in the output variable $j$ as a consequence of either a 10% increase or 10% decrease ($\Delta_{pi}$) in the parameter value.

## Statistics and reproducibility

For single-molecule fluorescence experiments the analyses constitute curve-fitting described in full above. No statistical hypothesis testing was performed but non-overlapping 95% confidence intervals are assumed to correspond to statistically significant differences. For high-speed AFM data, histograms were fitted by one or two Gaussians. The meaning, and origin, of central measures (arithmetic mean values) and error bars are described in the figure legends.

No sample size calculations were performed prior to the experiments. However, appropriate sample sizes for single-molecule fluorescence experiments have been determined previously and found to be in the range of 100 and 1000 single-molecule events with limited effects of the sample size within this range on the quantitative characteristics of the resulting distributions[35]. Note that sample sizes for single-molecule fluorescence at the highest [Pi] levels are necessarily small as a direct effect of Pi on the Alexa647-ATP binding probability.

For single-molecule fluorescence experiments, data were only excluded based on the strictly applied criteria described in full detail above. For hs-AFM experiments, no full data sets were excluded. Only a few images in the successive hs-AFM sets were omitted from data analysis due to blurriness or unclearness.

In the replication of single-molecule fluorescence experiments, each binding event of a fluorescent ATP molecule to a myosin molecule was assumed to be an independent random event independent of the experimental occasion, or myosin batch as supported by previous findings of very similar kinetic properties of myosin from different preparations[76,102]. Furthermore, the assumption is supported by similar results in experiments performed on different days and/or using different myosin preparations (Fig. 2b and other work[35]) and no consistent changes in behavior during the course of a given experiment[34]. For hs-AFM, the experiments with F-actin-HMM complexes were replicated multiple times using different experimental conditions, including the absence of any nucleotides (rigor), presence of different ADP concentrations as well as complexes with metavanadate and para-aminoblebbistatin at desired concentrations. In the experiments (with ADP at the myosin active site), we assumed each image frame of an actin-attached myosin molecule as one independent sample from a single distribution of actin-attached myosin head conformations in equilibrium, unaffected by experimental occasion and/or myosin preparation per se but only by the experimental condition (e.g., ADP alone, ADP + PAB, etc.). This is justified as follows. First, myosin heads are believed[42,47] to be independent force-generators, meaning that the attachment of one head does not affect the conformation of another head (assumption corroborated in Supplementary Fig. 16a). Second, the independence between experimental occasions and myosin batch is supported by our data in Supplementary Figs. 16, 18 as well as

previous data showing very similar kinetic properties of myosin from different preparations[76,102]. Accordingly, we found similar results in three hs-AFM experiments performed on different days using different surface preparations, solutions and HMM from two different myosin preparations. This is also consistent with previous results showing negligible changes in function over prolonged experimental periods[35,103]. Third, and finally, the ergodicity principle allows distributions of myosin head conformations (e.g., the distributions shown in Fig. 6 and Supplementary Figs. 16, 18) to be obtained either from a time series or an ensemble of myosin conformations or a combination, as used here. The quantitative parameter estimates are then obtained by fitting Gaussian functions or double-Gaussian functions to these distributions (based on >200 data points) rather than using small sample based estimates of mean values and errors. This procedure also circumvents the need for sample size calculations as the entire distributions are considered.

Statistical analyses, e.g., two-sided statistical hypothesis tests and curve-fitting were performed using Graph Pad Prism software v. 8 – 9.2 (Graph Pad Software LLC).

For molecular and mechanokinetic modeling, issues related to statistics and reproducibility are not applicable because no stochastic, e.g., Monte-Carlo simulations were performed. That is, all results of the modeling computations lack uncertainty and are without variability between independent runs.

### Reporting summary

Further information on research design is available in the Nature Research Reporting Summary linked to this article.

### Data availability

Data from modeling are available in the main paper (Figs. 1, 4, 5), Excel source data files for Figs. 4, 5 and the Supporting information (Supplementary Figs. 1–11 and Supplementary Tables 3–7). Data from single molecule experiments are available in the main paper (Figs. 2, 3), in Excel source data files for Figs. 2.3 and in Supplementary Tables 1, 2. Raw data (movies) from single-molecule fluorescence experiments are provided upon reasonable request or through Zenodo (https://doi.org/10.5281/zenodo.6793540). The high-speed AFM data (raw data and processed data) are included in Supplementary Material (Supplementary Figs. 12–16, 18, 19, Supplementary Movies 2–17, and the Excel source data file for hs-AFM data). Throughout the manuscript, we have used deposited myosin structures with the following accession codes: 5N6A (Pre-powerstroke); 4PFO (Pi-release state); 3I5F (MgADP myosin S1); 1QVI (Pre-powerstroke); 6Z7U (with Blebbistatin). Source data are provided with this paper.

### Code availability

The code of the Matlab routines used for single-molecule analysis are available upon reasonable request or through Zenodo (https://doi.org/10.5281/zenodo.6793540). The code for mechanokinetic modeling is provided in the Supplementary Materials of the paper. The code of the pixel-search software used for AFM image analysis can be accessed at https://elifesciences.org/articles/04806. The molecular modeling software used here are all available free online.

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

## Acknowledgements

We acknowledge funding from The Swedish Research Council (grant number 2019-03456), The Faculty of Health and Life Sciences at Lin-naeus University, Sweden and The Natural Science and Engineering Research Council of Canada (grant number RGPIN-2016-05317). The computations were enabled by resources provided by the Swedish National Infrastructure for Computing (SNIC) at the PDC center, partially

funded by the Swedish Research Council through grant agreement no. 2018-05973. Finally, we gratefully acknowledge valuable discussions with Professors Henry Hess and Sven Tågerud.

## Author contributions

A.M. conceived the study. A.M., R.F., D.E.R., L.M., M.U., and O.M. designed different experiments and modeling studies. A.M. performed mechanokinetic modeling. R.F. and L.M. performed molecular modeling. M.U., with the help of L.M. performed and analysed single-molecule experiments. O.M. performed hs-AFM experiments that were analysed together with D.E.R. A.M. wrote the paper with extensive input from all co-authors. All authors approved the final version of the manuscript before submission.

## Funding

## Competing interests

The authors declare no competing interests.
