## [Peer Review File · Nature Communications]

REVIEWER COMMENTS

Reviewer #1 (Remarks to the Author):

The present study explores the idea of 'secondary binding sites' during the release of Pi through the molecular backdoor tunnel of the myosin proteins. This idea comes from previous experiments found in the literature, as those of the laboratory of Davis O. Thomas using FRET measurements. With this idea in mind, the present study proposes a new 'model of description of differential equations' of the mechanism of action of the myosin proteins and their generation of mechanical power from chemical stored energy. The main novelty of the study, their working hypothesis, is that during the cycle of activity of myosin, the process of release of the Pi molecule from the myosin, relative to the timing of the power-stroke, could be as follows: 1. The Pi molecule leaves the active site before the power-stroke; 2. The Pi molecule travels through the myosin molecule through a path named the 'back door' where it temporally binds to several secondary active sites; and 3. The Pi gets out of the myosin through the exit of the back door after the power-stroke is finished. To provide further support than the previous literature to their working hypothesis, the authors perform first the structural analysis using the Protein Data Bank data on the myosin II and myosin VI and point to the presence of a back door and secondary binding sites for Pi on myosin II. Next, the authors carry out the kinetic calculations for their working hypothesis using the foundations laid on a previous manuscript of the same authors (ref 19 of the manuscript); the authors claim that their kinetic calculations provide solve the contradictions of the earlier literature. The experimental data that the authors provide is twofold: the authors check the validity of their working hypothesis using the dwell time events of fluorescence-tagged molecule Alexa647-ATP in competitive assays against Pi and the ADP-vanadate complex during the myosin II cycle (they use a recently published optical fluorescence procedure from the same authors, ref 32), in addition, the authors measure the angle of the lever arm of myosin II using High-Speed AFM, under ADP, ADP-vanadate complex and ADP with blebbistatin. In the case of the fluorescence data, the dwell lifetimes are fitted using three dwell times, from this fitting they assign the interaction of Pi and Alexa647-ATP with the myosin protein to three categories of bindings: fast ($3s^{-1}$) unspecific binding, slow ($0.3s^{-1}$) unspecific binding, and 'very slow' ATPase of myosin binding ($0.05s^{-1}$). The authors introduce ADP-vanadate in the imaging solution, they observe that the 'very slow' binding events disappear, based on this fact the authors impute the 'very slow' binding to the binding to and process of the ATPase of myosin. The authors impute the 'slow' and 'fast' binding events to the passage of Pi and Alexa647-ATP through the back door tunnel, for it they exclude that the 'slow' and 'fast' binding are due to the binding to the external surface of the myosin, which they calculate must be faster than $150s^{-1}$. In the last section of the manuscript, the authors state that their calculations state that only if the following condition is met, their model is valid (latter they comment that other model in the literature also meets the same condition, for myosin II ref 29, and for myosin V ref 48); the presence of ADP at the active site is sufficient condition for the power-stroke to take place. They provide observations by HS-AFM of the angle between the myosin arm and the actin filament that show that power-stroke takes place with ADP. In the presence of ADP, the myosin heads show an angle (40° - 50°) with respect to the actin filament; consistent with a molecular state posterior to the power-stroke (rigor state). Next, the authors show that the addition of vanadate that blocks the ADP in the active site causes the myosin to

adopt most often the pre-stroke configuration whose angle with the actin filament is higher than 50°. Finally, blebbistatin that favors the pre-stroke configuration by binding outside the active site is incorporated into the imaging buffer, it is found that the angle with the actin filament gets higher than 50° as for vanadate. Under all the studied conditions, the authors visualize —thanks to the molecular-level resolution provided by the HS-AFM— that the myosin switches back and forward between the pre-powerstroke and post-powerstroke structural states, importantly for their conclusions, the authors observe that there is more structural switching when the myosin is exposed to ADP plus vanadate than when it is to ADP alone. importantly, they interpret this finding as a testimony of the motion of the vanadate between the active site of myosin and secondary binding sites along with the back door channel, as expected in their model, which they put forward as proof for their model.

This study contains a very interesting hypothesis and some nice data. Nevertheless, I cannot support the publication of this work in the present form. The reasons are stated next:

Major issues.

1) The authors claim data from other authors (as refs 9 and 26) provide the main proof for their hypothesis, in this regard, the experimental data in the manuscript is not so significant.

2) The high-speed atomic force microscopy dataset and analysis are not reliable.

2.a) Using hs-afm movies the authors obtain hundreds of data points of the angle between the myosin head and the actin filament under three different biochemical conditions. These hs-afm movies are not shown. The authors only show us a small fraction of the totality of their hs-afm footage. It is to expect that such a small fraction of the movie corresponds to the movie frames of the best quality of image. Yet, in the small fraction of the totality of their hs-afm footage that the authors show us, the quality of the image of most of the frames is insufficient for a clear determination of the angle between the myosin head and the actin filament. It is necessary, for the sake of transparency, that the authors show the totality of their hs-afm footage used to measure the angles, and that they in which frames they performed the measurements.

2.b) Figure S12, the authors provide two measurements of the range of variation of the angle of myosin head with respect to the filament under two conditions, ADP-vanadate (100 μ M) and ADP-PAB (10 μ M), they measure angles of 33° and 32°, only 1° of difference! The uncertainty of such measurement is not provided. It is hard to understand how can the authors detect an angle difference of only 1° using their hs-afm movies that are, at least the ones shown, of low quality of the image. The consistency of the data analysis of the hs-afm movies should be proved in more detail.

2.c) The hs-afm movies show that many myosin molecules are attached to two actin filaments simultaneously, instead of one. The authors did not comment on this issue which can be significant for the angles they measure. The relative position of the binding sites of the myosin heads on each of the two actin filaments could influence the angle of each of the two heads; that in the videos seems to pull from each other. The authors should address this issue by, for example, a comparison of the angles on myosin proteins with their heads attached to one and two filaments.

2.d) The hs-afm data shows that the myosin fluctuates between the pre-powerstroke and the post-powerstroke states. The authors claim that when the myosin is exposed to ADP and vanadate there is more fluctuation than when the myosin is exposed to only ADP (without vanadate). The authors use this claim as evidence of support for their model, nevertheless, there could be a flaw in this observation. The claim of the authors comes from the data of the hs-afm movies S2 (presence of ADP) and S3 (presence of ADP+vanadate). One important aspect the authors do not take into consideration is that the imaging parameters of the S2 and S3 movies are not equal: the tip scanning speed is 19nm/ms at the movie S2, whereas it is 64nm/ms at S3. Hence, the tip motion was 3.37 times faster during the acquisition of the S3 movie than during the acquisition of the S2 movie. 3.37 times is a significant speed difference, it could be at the origin of the higher rate of fluctuation of myosin found in the S3 movie. Because the tip motion was faster in S3, the error in the sample contouring, the dissipated energy, the impact with the sample, was larger in the S3 movie than in the S2 movie. Thus, it is the case that the set of hs-afm data presented in the study cannot elucidate of whether it was the incorporation of vanadate in the imaging solution of S3 or the faster imaging speed of S3 which was at the origin of the higher rate of fluctuations of the myosin observed at S3. Further data are required to elucidate this issue, hs-afm data acquired at the same imaging speed would be welcomed, moreover, it would also be interesting to test the independence of the pre-powerstroke to post-powerstroke fluctuation with respect to the hs-afm imaging speed, for it, different imaging speeds should be tested under vanadate and ADP conditions and, next, the fluctuation rate should be plotted against the imaging speed.

Minor issues.

1) The authors claim that their new model solves the inconsistencies of previous models and experimental data. Nevertheless, in its present form, the manuscript does not provide sufficient details and clarity on which are these points of inconsistency from the previous literature. This is a critical aspect that needs more explanation, the reader should understand better the facts that drive the inconsistencies of the previous models and experiments.

2) In their structural analysis, the authors omitted the possibility found by other structural studies (<https://doi.org/10.1021/acs.jpcc.0c10004>) that, instead of a single pathway, several gateways are

present for the exit of Pi. If such circumstance is included, it is possible that the predictions of their model for the functioning of the myosin could change. Please discuss this issue.

3) The manuscript places high relevance to the delay of the release of Pi from the myosin into the solution. Nevertheless, the manuscript does not provide information on how long this delay is to be expected. Please detail this point.

4) As the manuscript is structured, there is a lot of important information placed in the figure captions that are missing from the main text. To facilitate the reading, the distribution of the information should be more equilibrated.

Reviewer #2 (Remarks to the Author):

A new model is proposed to explain the relationship between phosphate release and force generation in skeletal muscle myosin, which is a crucial for understanding energy transduction in myosins and relevant to other mechanoenzymes. The key feature of this model is that Pi can bind to an alternate site on myosin and thus Pi-release occurs in two phases – Pi leaving the active site and Pi release from 2ndary binding site. This model is similar to what was proposed by Sweeney and Houdusse based on x-ray crystallography studies of myosin VI. The authors provide experimental support for their hypothesis using single molecule binding events with fluorescent ATP in the presence of varying Pi concentrations. They also use high-speed atomic force microscopy to demonstrate the lever arm rotation can occur with ADP in the active site. In addition, a mechanokinetic model was developed to test their hypothesis. In their model, the alternate Pi binding site does not alter the myosin power stroke making the single molecule mechanical experiments independent of free Pi concentrations. Pi binding to the second site does reduce isometric force without altering contractile velocity.

This is a very interesting study that provides new insight into a long standing question in the molecular motors field. The new model is very exciting and mostly well explained in the article and the supplementary information. However, the experimental support for the secondary Pi binding sites has some weaknesses. Also, in my opinion the high speed AFM results are over interpreted.

(Figure 1) The molecular modeling is very impressive and convincingly demonstrates the possibility of a secondary Pi binding site in myosin II.

(Figure 2) Specific comments on single molecule ATP binding events in the presence of Pi. The authors observed a reduction in the number of fluorescence spots with increasing Pi, suggesting competition between Pi and the fluorescent ATP for binding to myosin. This could be because they both compete for

the active site. It has been found previously that Pi can compete with ATP for the active site (Amrute-Nayak et al. 2008)(Debold et al. 2011). The authors go on to demonstrate several different dwell times whose amplitudes were decreased by Pi, consistent with different binding sites for Pi that alter the period of time the fluorescent nucleotide is bound. However, another interpretation of these results is that the fluorescent ATP can bind specifically and go through an ATPase cycle, non-specifically and rapidly detach, or bind to the active site but detach prematurely before going through an ATPase cycle. The myosin can also be in different conformations that could explain the different ATP binding/dissociation rates. The overall number of spots is dramatically reduced because of the ability of Pi to compete for the active site, which explains the most striking result.

(Figure 3) The authors use ADP.vanadate to block the active site and address my concern above about competition for the active site. They find that the number of spots is reduced dramatically, which is expected since all or most active sites are blocked. The binding events that they do observe are similar to Figure 2. However, vanadate is not perfect and there could be some active sites that are free. The authors could verify experimentally that under the conditions of the experiment all active sites contain ADP.vanadate (blocked) using ATPase measurements. Overall, the number of spots is reduced because most of the active sites are blocked but those that are open behave the same as Figure 2. Pi can still compete for the active site as stated above.

(Figure 4) In the kinetic model is unclear how the presence of excess Pi alters specific reaction steps. This is important because the data demonstrate that the isometric force is sensitive to Pi but the shortening velocity is not. They also demonstrate that Pi does not alter the power stroke. This is exciting because the model agrees with experimental data but it could be explained better in the text the key steps that are Pi-sensitive and how changes in specific reaction steps lead to the outputs (Figure 4, b-d).

(Figure 6) The high speed AFM was performed to demonstrate a power stroke can still occur with only ADP in the active site. Since the model proposed required Pi-release from the active site first, the power stroke should occur with ADP in the active site. It is known that there is an additional swing of the lever arm with ADP release, although typically not detectable in skeletal muscle myosin, which can complicate the interpretation of experiments. The authors demonstrate mainly one conformation of the lever arm in ADP and another conformation in ADP-vanadate. There are two conformation in the presence of the drug blebbistatin which traps ADP in the active site. This is quite interesting and suggests the drug allows the lever arm to adopt two different conformations. However, it is unclear if the results demonstrate the movement of the lever arm from the pre to the post-power stroke conformation. For example, is it clear that myosin is remaining in a strongly bound conformation during the transition? The traces look like in all three conditions the lever arm can adopt a pre or post-power stroke. The data emphasizes the different conformations of the lever arm that are possible in each condition but does not emphasize specific events that demonstrate a transition from the pre- to post-power stroke state.

Reviewer #3 (Remarks to the Author):

This manuscript details a combined experimental and modeling approach to determining the detailed kinetics of Pi release during the motor activity of myosin. The authors show that in certain isoforms of myosin, there is a "back door" binding site for the Pi. Using experiments and modeling, the authors determined the rate constants for binding of Pi, and showed that there are multiple rate constants, consistent with multiple binding sites. Using this analysis, the authors created a revised model for the power stroke kinetics of myosin that explain a number of aspects that could not be explained previously, most notably the Pi independence of the power stroke rate.

Overall, the results presented here provides a more detailed and quantitative model for the myosin-actin interaction. It can explain some aspects relevant to disease, such as why certain mutations are pathological. I am not an expert on the experimental approach, but it seemed fine. The modeling work was sound and supported the overall conclusions that were determined.

While this work will definitely impact the molecular motor community, its major weakness is that the results may not provide significant impact to fields outside this fairly narrow community.

Reviewer #4 (Remarks to the Author):

I congratulate the authors for this highly interesting manuscript. The idea of unifying conflicting perspectives of power stroke and Pi release coupling by implementing a second binding site of the cleaved Pi in the kinetic scheme is ingenious and strongly supported in my feeling by the high-speed AFM structures of myosin presented in the paper. As I was requested by the editor to particularly review the kinetic models in the manuscript, I have only one main comment addressed to the author(s) performing the model computation. That comment is, however, really important because addressing it will greatly increase the significance of the models in terms of both, reliability of rate constants and consistency with thermodynamics.

The models in the manuscript consisting of branched pathways do not account for thermodynamics. When models implement different pathways leading from the same initial state before branching to the same final state after the branches, the values of rate and equilibrium constants within the branched part of the model cannot freely selected, some of them have to be adjusted accordingly by the principle of detailed balance to fulfil thermodynamics. I am almost sure that the expert performing the simulations is aware of this. However, in this regard, the transition in Fig. 4a leading from the AMDP'H to the AM/AMD state is superficial and inconsistent with thermodynamic principles. If it would not exist and power stroke transitions and second release steps of Pi are the same in the two diverging pathways as in this manuscript, the model would be consistent.

Similarly, in Fig S8a and b, the merging transitions from different initial states to the same final ATP state (designated with the same rate constant and computed by using the same values for k_{det} in the computations) are also inconsistent with thermodynamics. There should be only one transition in each model leading to the ATP state, i.e., the transition from the ADP state. The other transitions designed by this merging arrow are superficial and already covered by the forward transitions leading to the ADP state in the model and their inclusion contradicts thermodynamics.

Thus, I suggest correcting the model schemes in Fig. 4 and Fig. S8 by deleting the superficial transitions.

Consequently, for making models correct, delete in the model equations the superficial $-k_{det} \cdot a_{XX}$ -terms in the differential equations (where a_{XX} are other states than the ADP state), re-run the corrected models and re-plot the new values obtained by the simulations in the figure templates.

Minor comments:

Fig. S8 caption: "On the assumption that $k_{det}=0$ the isometric steady-state force as a function of $[Pi]$ is given by the expression in the box (with constants c_1 and c_2 given to the left). This expression gives a force- $[Pi]$ relationship that is negligibly different from that obtained by numerical solution of the differential equations with $k_{det}=5 \text{ s}^{-1}$ (plotted in main Fig. 4c)."

Does this also account for the corrected model?

Generally, the literature is well cited and balanced in the manuscript. I would like to point to studies of Lionne et al (1995) FEBS Letters with myofibrils and He et al (1997) J Physiol with muscle fibres that have demonstrated that there is no burst of free Pi after initiating the cross-bridge ATPase cycle and muscle fibres. The surprising absence of free Pi bursts in these studies provides straightforward evidence for delayed release of phosphate by working cross-bridges addressed in the manuscript.

Yours, sincerely

Robert Stehle, PhD

Dear Reviewers,

We are grateful for a careful and constructive reviews of our manuscript that have helped us to achieve major improvements. Please find below, our detailed point to point response to the reviewer comments with the original comments in italics. In addition, the key changes are indicated by **track changes** in submitted versions of the manuscript and Supplementary Information for review only. A list of references that we cite below, are located after the responses to all reviewers. Please note, that Figures 4 and 6 in the main paper have been partly (Fig. 4, panel a) or extensively (Fig. 6) changed without indication in the track changes version of the manuscript in order not to hamper readability. Finally, we have carefully gone through the paper with respect to typographical, formal, grammatical or factual errors. We have also made some minor changes to ensure consistency and increase clarity. The resulting corrections are not always indicated by track changes if they just concern one or a few words. Page numbers etc. refer to the version of the manuscript **with** track changes.

Response to Reviewer #1 (Please note, page numbers etc. refer to the version of the manuscript **with** track changes)

The present study explores the idea of 'secondary binding sites' during the release of Pi through the molecular backdoor tunnel of the myosin proteins. This idea comes from previous experiments found in the literature, as those of the laboratory of Davis O. Thomas using FRET measurements. With this idea in mind, the present study proposes a new 'model of description of differential equations' of the mechanism of action of the myosin proteins and their generation of mechanical power from chemical stored energy. The main novelty of the study, their working hypothesis, is that during the cycle of activity of myosin, the process of release of the Pi molecule from the myosin, relative to the timing of the power-stroke, could be as follows: 1. The Pi molecule leaves the active site before the power-stroke; 2. The Pi molecule travels through the myosin molecule through a path named the 'back door' where it temporally binds to several secondary active sites; and 3. The Pi gets out of the myosin through the exit of the back door after the power-stroke is finished. To provide further support than the previous literature to their working hypothesis, the authors perform first the structural analysis using the Protein Data Bank data on the myosin II and myosin VI and point to the presence of a back door and secondary binding sites for Pi on myosin II. Next, the authors carry out the kinetic calculations for their working hypothesis using the foundations laid on a previous manuscript of the same authors (ref 19 of the manuscript); the authors claim that their kinetic calculations provide solve the contradictions of the earlier literature. The experimental data that the authors provide is twofold: the authors check the validity of their working hypothesis using the dwell time events of fluorescence-tagged molecule Alexa647-ATP in competitive assays against Pi and the ADP-vanadate complex during the myosin II cycle (they use a recently published optical fluorescence procedure from the same authors, ref 32), in addition, the authors measure the angle of the lever arm of myosin II using High-Speed AFM, under ADP, ADP-vanadate complex and ADP with

blebbistatin. In the case of the fluorescence data, the dwell lifetimes are fitted using three dwell times, from this fitting they assign the interaction of Pi and Alexa647-ATP with the myosin protein to three categories of bindings: fast (3s-1) unspecific binding, slow (0.3s-1) unspecific binding, and 'very slow' ATPase of myosin binding (0.05s-1). The authors introduce ADP-vanadate in the imaging solution, they observe that the 'very slow' binding events disappear, based on this fact the authors impute the 'very slow' binding to the binding to and process of the ATPase of myosin. The authors impute the 'slow' and 'fast' binding events to the passage of Pi and Alexa647-ATP through the back door tunnel, for it they exclude that the 'slow' and 'fast' binding are due to the binding to the external surface of the myosin, which they calculate must be faster than 150s-1. In the last section of the manuscript, the authors state that their calculations state that only if the following condition is met, their model is valid (latter they comment that other model in the literature also meets the same condition, for myosin II ref 29, and for myosin V ref 48); the presence of ADP at the active site is sufficient condition for the power-stroke to take place. They provide observations by HS-AFM of the angle between the myosin arm and the actin filament that show that power-stroke takes place with ADP. In the presence of ADP, the myosin heads show an angle (40°-50°) with respect to the actin filament; consistent with a molecular state posterior to the power-stroke (rigor state). Next, the authors show that the addition of vanadate that blocks the ADP in the active site causes the myosin to adopt most often the pre-stroke configuration whose angle with the actin filament is higher than 50°. Finally, blebbistatin that favors the pre-stroke configuration by binding outside the active site is incorporated into the imaging buffer, it is found that the angle with the actin filament gets higher than 50° as for vanadate. Under all the studied conditions, the authors visualize —thanks to the molecular-level resolution provided by the HS-AFM— that the myosin switches back and forward between the pre-powerstroke and post-powerstroke structural states, importantly for their conclusions, the authors observe that there is more structural switching when the myosin is exposed to ADP plus vanadate than when it is to ADP alone. importantly, they interpret this finding as a testimony of the motion of the vanadate between the active site of myosin and secondary binding sites along with the back door channel, as expected in their model, which they put forward as proof for their model.

This study contains a very interesting hypothesis and some nice data. Nevertheless, I cannot support the publication of this work in the present form. The reasons are stated next:

Response: We thank the reviewer for careful reading and analysis of the manuscript.

Major issues.

1) The authors claim data from other authors (as refs 9 and 26) provide the main proof for their hypothesis, in this regard, the experimental data in the manuscript is not so significant.

Response: We have possibly been unclear, because refs 9 and 26 (now 10 and 27, used below), in isolation do not provide any evidence for our hypothesis. Several papers like these suggest that Pi-release into solution occurs **after** the power-stroke, but maybe this is not what the reviewer actually means here?

Anyway, in these papers (including refs 10 and 27) the data are not viewed in a context with other results. For instance, as we point out e.g. in the Introduction of our manuscript, the virtually constant sliding velocity and monotonous decrease in isometric force with increased [Pi] are not consistent with the results in refs 10 and 27 (and similar papers) if previously published models are used. Therefore, the evidence for, and formulation of, **one** unifying model that explains all (at least most) experimental findings is a key new result of our study. Previously, different models have instead been required (e.g. see p 2, 2nd paragraph – p3, 1st paragraph, p 11, last paragraph, p. 12 first paragraph and Figs. S5-S6) to explain different effects of altered [Pi]. Therefore, the model to explain all conflicting results within one framework is, as far as we can judge, a major accomplishment that is highly significant in its own right. The fact that it is partly tested against previously published results for contractile function does not take away this fact. On the other hand, importantly, our paper also provides **new** experimental findings to support the model, both based on the studies of Pi-binding and the high-speed AFM data. We have elaborated extensively (see details below) on the experimental data in the present revision with the hope to make the results clearer and more convincing. On this basis we now believe that they more strongly support our model than in the previous version of the manuscript. Finally, we think that the issue raised here is related to the minor issue 1, also raised by this reviewer. Therefore, the changes made in response to that point are also of relevance for clarifying this major issue resulting in the changes specified below.

Changes made (Major issue 1): The changes include modifications in: i. the Introduction (on p 2, 2nd paragraph and p3, 1st paragraph), ii. the Results (p 11-12) and iii. the Discussion (p 19, 1st paragraph).

2) *The high-speed atomic force microscopy dataset and analysis are not reliable.*

2.a) *Using hs-afm movies the authors obtain hundreds of data points of the angle between the myosin head and the actin filament under three different biochemical conditions. These hs-afm movies are not shown. The authors only show us a small fraction of the totality of their hs-afm footage. It is to expect that such a small fraction of the movie corresponds to the movie frames of the best quality of image. Yet, in the small fraction of the totality of their hs-afm footage that the authors show us, the quality of the image of most of the frames is insufficient for a clear determination of the angle between the myosin head and the actin filament. It is necessary, for the sake of transparency, that the authors show the totality of their hs-afm footage used to measure the angles, and that they in which frames they performed the measurements.*

Response: We agree with the reviewer that it is difficult to clearly observe the lever arm, as it is short (~8-12 nm), thin and highly dynamic, particularly in the presence of ADP-metavanadate and ADP-para-aminobebistatin. We have therefore used several approaches to aid our estimates of lever arm

orientation. First, capturing hs-AFM movies where two heads are simultaneously attached to two different actin filaments under weak or strong binding conditions facilitates the detection of the lever arm and its dynamic movements as argued in a previous study¹. Further, during the course of our analysis, we compared the hs-AFM images of the actin-HMM complex to lever arm orientations in actin-myosin atomic models, S1 structures and simulated hs-AFM records in the presence of ADP, ADP-Vanadate or ADP-blebbistatin. Myosin heads orientation, actin polarity and center of mass values of the myosin heads were then used to measure the lever arm angles (see Movies S2-S7 cited in the main text and all other hs-AFM Movies S8-S17 as well as hs-AFM Source Data file (Excel filed)).

The transition between strong (post-power-stroke) and weak (pre-power-stroke) binding states and vice versa can generally be clearly observed, with $\sim 30^\circ$ difference in mean values of lever arm orientation from observed distributions (for ADP-metavanadate and ADP- para-aminoblebbistatin, Fig. 6 and Figs. S13-S14 and Movies S4-S7).

Despite our careful procedure as described above, we are aware that the exact lever arm orientation may be challenging to judge without any ambiguity. It is therefore important to note that some clear changes in orientation between pre- and post-power stroke states exist between frames e.g. as indicated in Fig. R4 (Fig. S15) and as can be found in frame-by-frame inspection of the movies, particularly Movies S5-S7. Examples are seen during the time periods 1.2 s-1.4 s in Movie S6 and 2.40-2.55-2.70-2.85-3.0 s as well as 3.45-3.65 s in Movie S7 (see also hs-AFM Source dataset (Excel) for angle measurements and new Movies S15-S17).

While we take the mentioned non-ambiguous events as particularly strong support for our claims one may, of course, argue that the observed transitions are due to detachment from the post-power-stroke state followed by a recovery stroke in the detached head with transition into the pre-power-stroke state and subsequent reattachment to actin in the pre-power-stroke state conformation. However, in our experimental approach in the absence of ATP the latter scenario is unlikely to occur as now mentioned on p. 16, final paragraph (just before Fig. 6).

Moreover, to improve the visibility of the myosin heads and the lever arms, we have now applied a 3D surface plot isoline filter (imageJ, v.1.47) to the images and movies presented in the supplementary information (Movies S2-S7, the new Figs. S12-S15 are also inserted below as Figs. R1-R4). These filtered hs-AFM movies have been combined with the original hs-AFM movies for a better correlation of the lever arm dynamics under all conditions investigated. The estimated lever arm position is highlighted to indicate how the angles were measured and where power-stroke transitions (or their reverse transition) occurred (see Movies S2-S7). We hope that these changes, together with other clarifications explained above, will allow also readers without previous experience from hs-AFM data to follow our arguments and trust our data.

In addition to the new data and Supporting movies cited in the text of the main paper (Movies S2-S7), we now also make all hs-AFM raw data movies

(Movies S8-S17) available in the Supporting information and some of these are used to provide data for Fig. 6 as now indicated in the legend of that figure.

Summary of changes made (Major issue 2a): We have quite extensively changed and updated Fig. 6 and the associated legend. We have also updated and expanded the supporting Movies directly cited in the text (Movies S2-S7) and combined them with 3D-surface plot isoline filtered versions of the movies to better visualize the myosin lever arm. We have also included additional movies S8-S17 for all studied conditions as well as a the hs-AFM Source Data file (Excel) reporting x,y, z coordinates, lever arm angles and drift analysis. Finally, we have modified the main text to better guide the reader, e.g. with regard to the importance of observing movies frame by frame (p 15, end of final paragraph and p 16 as well as legends of Supporting Movies). We have now also more openly exposed the challenges in interpreting hs-AFM data (p 15, end of final paragraph and p 16).

Fig. R1. New Fig. S12. ADP-lever arm position in the actin-myosin complex. a-b Atomic models and corresponding simulated hs-AFM images of the actin-myosin S1-ADP complex (PDB 2Y83, PDB 3I5F) for two experimental approaches used in this study. Two heads attached to a given actin filament in **a** and to different actin filaments in **b**. Simulated hs-AFM images were obtained by Bio-AFM viewer software (v.2.0). **c-d** The ADP-lever arm position of myosin relative to actin filament, when two HMM heads were bound to one actin filament in **c** and when each head was bound to a different, nearly parallel actin filaments in **d**. The ADP-lever arms and myosin heads are highlighted by the 3D-filtered images and the angle measurement of lever arm position relative to actin filament is indicated by green lines. The scale bars are 30 nm for non-zoomed images and 10 nm for zoomed images.

Fig. R2. New Fig. S13. ADP-vanadate lever arm position in the actin-myosin complex. **a-b** Atomic models and corresponding simulated hs-AFM images of the actin-myosin S1-ADP-vanadate complex (PDB 2Y83, PDB 1QVI) for two experimental approaches used in this study. Simulated hs-AFM images were obtained by Bio-AFM viewer software (v.2.0). **c-d** The ADP-vanadate lever arm position of myosin relative to actin filament, when two HMM heads were bound to one actin filament in **c** and when each head was bound to the two parallel actin filaments in **d**. Zoomed images to the right in each panel. The ADP-vanadate lever arms and myosin heads are highlighted by the 3D-filtered images and the angle measurements of lever arm positions relative to actin filament are indicated by green lines. The scale bars are 30 nm for non-zoomed images and 10 nm for zoomed images.

Fig. R3. New Fig. S14. ADP-blebbistatin lever arm position in the actin-myosin complex. **a-b** Atomic models and corresponding simulated hs-AFM images of the actin-myosin S1-blebbistatin complex (PDB 2Y83, PDB 6Z7U) in the pre-power stroke state. Simulated hs-AFM images were obtained by Bio-AFM viewer software (v.2.0). **c** The difference in lever arm position during the power stroke comparing pre-power-stroke position (PPS) and post-power-stroke position (POST). **d** hs-AFM snapshots of myosin heads binding

to the two parallel actin filaments in pre-power-stroke positions with zoom in of area within box to the right. The ADP-blebbistatin lever arms and myosin heads are highlighted by the 3D-filtered images and the angle measurement of lever arm position relative to actin filament is indicated by green lines. The scale bars are 30 nm for non-zoomed images and 10 nm for zoomed images.

Fig. R4. New Fig S15. Measurement of the lever arm position in the actin-myosin complex. **a** hs-AFM successive images of actin-myosin-ADP-vanadate complex with myosin heads bound to two parallel actin filaments and **b, c** corresponding lever arm and heads orientations used to calculate the angle of the lever arm. The transition from the strong, post-power-stroke (0.15 s), to the weak pre-power-stroke (0.30-0.45 s) lever arm positions of the lower myosin head is depicted. Scanning rate 6.7 fps for a scan area of 150 x 120 nm² with 80 x 64 pixels. Scale bar 30 nm.

2.b) Figure S12, the authors provide two measurements of the range of variation of the angle of myosin head with respect to the filament under two conditions, ADP-vanadate (100 μ M) and ADP-PAB (10 μ M), they measure angles of 33° and 32°, only 1° of difference! The uncertainty of such measurement is not provided. It is hard to understand how can the authors detect an angle difference of only 1° using their hs-afm movies that are, at least the ones shown, of low quality of the image. The consistency of the data analysis of the hs-afm movies should be proved in more detail.

Response: The position of the myosin lever arm in all studied conditions was calculated and plotted to observe the mean values from Gaussian distributions. For actin-myosin-ADP complex we observed a normal distribution with the mean angle of the lever arm corresponding to ~50° (Fig.6d, Fig. S12 as well as the new Figs. S16, S18). For the actin-myosin-ADP-metavanadate and actin-myosin-ADP- para-aminoblebbistatin complexes, the observed distributions were best fitted by the sum of two Gaussians with two peaks. The mean values for these peaks, obtained from the Gaussian distributions, are given in the legend of Figure 6. The first peak correlated with the lever arm orientation at the strong-binding state in the presence of ADP (53.6° for the ADP- para-aminoblebbistatin and 68.6° for the ADP-vanadate). The second

peak correlated with the lever arm orientation related to the weak binding state (87.7° for the ADP- para-aminoblebbistatin and 99.2° for the ADP-vanadate). The values mentioned by the reviewer represent the averaged difference in the lever arm position between the strongly-bound post-power-stroke and the weakly bound pre-power-stroke states. For ADP-vanadate this gave a difference (with error estimate) of $99.2 \pm 6.7^\circ - 68.6 \pm 13.8^\circ = 30.6 \pm 15.3^\circ$ and for ADP- para-aminoblebbistatin: $87.7 \pm 16.6^\circ - 53.6 \pm 8.3^\circ = 34.1 \pm 18.6^\circ$, respectively. These error estimates are now also given in the legend of Fig. 6 to clearly indicate that there is no statistically significant difference.

The lever arm positions calculated as the difference between averaged values of the pre- and post-power-stroke states in ADP-metavanadate and ADP- para-aminoblebbistatin conditions are indeed quite similar. However, when we presented the data as a time course (Fig. 6b) one can see that the fraction of frames with lever arm angle above $80-100^\circ$ is higher under ADP-blebbistatin than ADP-vanadate conditions.

Summary of changes made (major issue 2b): We have now clarified in the legend of Fig. 6 that we consider distributions of angular orientations with uncertainty in the mean values now given as standard deviations.

2.c) The hs-afm movies show that many myosin molecules are attached to two actin filaments simultaneously, instead of one. The authors did not comment on this issue which can be significant for the angles they measure. The relative position of the binding sites of the myosin heads on each of the two actin filaments could influence the angle of each of the two heads; that in the videos seems to pull from each other. The authors should address this issue by, for example, a comparison of the angles on myosin proteins with their heads attached to one and two filaments.

Response: The challenge associated with clear visualization of the lever arm and measuring its dynamic position is one of the reasons we decided to use the experimental approach with two HMM heads simultaneously bound between two parallel actin filaments. This approach, which was suggested in our recently published study¹, helped us to achieve higher spatial resolution with a visible lever arm in each myosin head in the presence of ATP. Another reason for using this approach is that it facilitates non-ambiguous study of the lever arm movement in a given molecule through successive hs-AFM frames. When HMM heads are bound to one actin filament, heads can go through attachment-detachment cycles while possibly changing position on the actin filament. A given binding site could be occupied by other myosin molecules, especially under weak binding conditions (presence of ADP-vanadate, ADP-blebbistatin or ATP). Thus, observation of myosin head binding to only one actin filament introduces uncertainties in the analysis of the lever arm movement.

As can be seen in the new provided movies, we were able to observe and measure the lever arm orientation also when HMM heads bound to one actin filament under different experimental conditions. The observed results revealed very similar angle distributions as for the case with binding of the two HMM heads to two actin filaments (New Fig.S16a-b, new Movie S10 (3rd

panel), Movie S12 and Movie S16). This is consistent with previous data for actin-myosin interactions in the presence of ATP¹.

In the presence of ADP, the lever arm angle relative to actin filaments, where each of the HMM heads simultaneously bound between parallel actin filaments, showed a mean value of $54.1^\circ \pm 12.1$ (SD). In the experiments, where the heads bound to one actin filament, the mean angle was $48.6^\circ \pm 13.6$ (SD). Neither for ADP-vanadate conditions did we observe any significant difference in the lever arm angle distributions between two experimental approaches with either one or two actin filaments (Fig. R5; New Fig.S16a). The strong binding state distribution peak was at $67.6^\circ \pm 13.8$ (SD) vs $54.1^\circ \pm 9.5$ (SD) for the heads simultaneously bound between parallel actin filaments and attached to one filament, respectively. The weak binding pre-power-stroke state peak revealed a higher mean value of $101.6^\circ \pm 7.7$ (SD) vs $79.4^\circ \pm 17.3$ (SD) for the heads simultaneously bound between parallel actin filaments and two heads attached to the one filament, respectively (New Fig.S16b). Importantly, however, an unpaired t-test showed no significant difference in lever arm angle between the two experimental situations with one and two actin filaments (ADP: $p=0.81$; ADP-vanadate: $p = 0.38$). These results are now added to the Supplementary Information (legend of Fig. S16) and mentioned in the Results section on p.16 (end of final paragraph) with reference to Fig. S16 (just before Fig. 6).

The reviewer also pointed out an interesting question: “*The relative position of the binding sites of the myosin heads on each of the two actin filaments could influence the angle of each of the two heads; that in the videos seems to pull from each other*”.

To clarify this issue, we measured the displacements of the myosin heads bound to the two actin filaments in successive hs-AFM frames. If we assume that the pulling of the heads is observed and it is mediated by lever arm movement, we need to test the conditions, where myosin lever arms are not expected to move. Further, we need to measure the head displacements over a time course to test for effect of one head pulling on the other. We performed an additional experiment for the heads in nucleotide-free state (rigor heads) simultaneously attached to two different actin filaments in the presence of 1 U/ml of apyrase to remove any traces of ADP or ATP contaminants in our buffers and in HMM preparations. The results are shown in Fig. R6 where two analyses were combined: i) analysis of the system drift during scanning and ii) analysis of the lower and upper head displacements calculated as a difference between the center of mass (COM) values of the head at the reference frame and the head COM values in the next frame. As one can see, there is no displacements or pulling of the heads by each other. Instead, there is a slight independent displacement of the heads due to the sample stage drift. In this experiment, the existing drift of the system, contributing to change the head position, is ~ 0.6 nm during the time course of the experiment. Thus, to avoid any uncertainty in our lever arm analysis, the stability of the offset and COM values in the subsequent frames during experiments was monitored for each frame used in analysis. The data for the drift and displacement analyses are included in the hs-AFM Source Data File and in a new Fig. S19. More details are given in the legend of the latter figure.

Summary of changes made (major issue 2c): We have now addressed the methodological issue related to studies being performed of the two myosin heads attached to nearly parallel actin filaments. First, we have motivated the approach by reference to a recent study¹. Second, we have performed new analyses comparing angle measurements in experiments using one and two actin filaments (New Fig.S16a-b, new Movies S10 (3rd panel), S12 and S16) and third, we have summarized the results of these analyses and the resulting arguments in new text in the main paper on p 16, end of final paragraph.

Fig. R5. New Fig. S16. The distributions of the lever arm angles obtained at different scan speeds and by using two experimental approaches for binding HMM heads to actin filaments. **a** lever arm distribution in the presence of ADP for the two types of HMM heads binding (number of events: 597 vs. 519); **b** lever arm distribution in the presence of ADP-vanadate for the two types of HMM heads binding (number of events: 504 vs. 190); ADP: $p=0.81$, unpaired t-test; ADP-meta-vanadate: $p = 0.38$, unpaired t-test for different between means of distributions (approximate normal distribution of the mean is assumed due to large n ; central limit theorem). **c** lever arm distribution in the presence of ADP for different frame rates (number of events: 659, 6.7 fps ($51.0^\circ \pm 13.7$), 268, 3.3 fps ($53.9^\circ \pm 11.8$), 189, 2 fps ($50.8^\circ \pm 12.2$)), $p = 0.93$ according to 1 way ANOVA. **d** lever arm distribution in the presence of ADP-vanadate for different frame rates (number of events: 504 vs. 82); $p = 0.92$, unpaired t-test. **e** Binding of HMM heads to two parallel actin filaments in the presence of ADP at 2 fps for a scan area of $200 \times 200 \text{ nm}^2$ with 120×120 pixels and 6.7 fps for a scan area of $150 \times 75 \text{ nm}^2$ with 80×40 pixels. Scale bars: 30 nm.

Fig. R6. New Fig. S19. The drift analysis of the system with x, y- center of mass (COM) position of lower and upper heads combined with the displacement analysis of each head in time. Note, that a decrease of the COM x-position after ~5 sec was due to manual change of the sample stage to center the HMM molecule bound to parallel actin filaments as can be seen in the Movie S11 (5th movie panel; contains five movies)

2.d) The *hs-afm* data shows that the myosin fluctuates between the pre-powerstroke and the post-powerstroke states. The authors claim that when the myosin is exposed to ADP and vanadate there is more fluctuation than when the myosin is exposed to only ADP (without vanadate). The authors use this claim as evidence of support for their model, nevertheless, there could be a flaw in this observation. The claim of the authors comes from the data of the *hs-afm* movies S2 (presence of ADP) and S3 (presence of ADP+vanadate). One important aspect the authors do not take into consideration is that the imaging parameters of the S2 and S3 movies are not equal: the tip scanning speed is 19nm/ms at the movie S2, whereas it is 64nm/ms at S3. Hence, the tip motion was 3.37 times faster during the acquisition of the S3 movie than during the acquisition of the S2 movie. 3.37 times is a significant speed difference, it could be at the origin of the higher rate of fluctuation of myosin found in the S3 movie than during the acquisition of the S2 movie. 3.37 times is a significant speed difference, it could be at the origin of the higher rate of fluctuation of myosin found in the S3 movie. Because the tip motion was faster in S3, the error in the sample contouring, the dissipated energy, the impact with the sample, was larger in the S3 movie than in the S2 movie. Thus, it is the case that the set of *hs-afm* data presented in the study cannot elucidate of whether it was the incorporation of vanadate in the imaging solution of S3 or the faster imaging speed of S3 which was at the origin of the higher rate of fluctuations of the myosin observed at S3. Further data are required to elucidate this issue,

hs-afm data acquired at the same imaging speed would be welcomed, moreover, it would also be interesting to test the independence of the pre-powerstroke to post-powerstroke fluctuation with respect to the hs-afm imaging speed, for it, different imaging speeds should be tested under vanadate and ADP conditions and, next, the fluctuation rate should be plotted against the imaging speed..

Response: We thank the reviewer for raising this issue. In our experiments the common frame rates were 3.3 fps and 6.7 fps. We have now added data obtained at different scan speeds in Fig. S16, also shown in Fig. R5 above. As can be seen from the new figure, the lever arm position under the ADP conditions at the different scan speed (2 fps, 3.3 fps, 6.7 fps) is quite similar, with the mean values from $\sim 51^\circ$ to 54° . These data are in agreement with the hs-AFM snapshots shown in the new figure. First, the movie was captured at the 2 fps and then the rate was increased to 6.7 fps, so that the same HMM molecules can be compared at two different frame rates (new Movie S9). Thus, for the strong binding state (presence of ADP), changing the frame rates in the range of 2 fps-3.3 fps-6.7 fps did not significantly change the distribution of the lever arm position relative to actin filament ($p = 0.93$, 1way ANOVA). In addition to the new Supporting figure and movies, this is now briefly mentioned in the main text on p 16, last line.

In case of ADP-vanadate, we observed some differences in the lever arm position (~ 11 - 14°) during slower scanning at 2 fps in comparison to faster rate at 6.7 fps (new Movie S14). The mean values of the first peak at the scan speed of 2 fps vs 6.7 fps were: $56.7^\circ \pm 10.2$ (SD) vs $67.6^\circ \pm 13.8$ (SD), respectively; the mean values of the second peak at the scan speed of 2 fps vs 6.7 fps were: $87.3^\circ \pm 22.5$ (SD) vs $101.6^\circ \pm 7.7$ (SD) (New Fig.S16; Fig. R5). Thus, for the weak binding state (presence of ADP-meta-vanadate), changing the frame rates from 2 to 6.7 fps did not significantly change the mean lever arm angle relative to the actin filament ($p = 0.92$, unpaired t-test)

Nevertheless, despite this evidence, the data in Figure 6 b, d (main text) were updated to use 6.7 fps scan rate for all the studied conditions.

Summary of changes made (major issue 2d): We performed additional experiments for binding HMM heads to actin filament at different frame rates and using two experimental approaches: i) binding two HMM heads to one actin filament or ii) simultaneously attaching each HMM head to two actin filaments. The comparison between these conditions showed slight, but not significant changes. Nevertheless, all analyses in Fig. 6 were updated to using only one scan rate of 6.7 fps. The results of the mentioned comparisons are reported in new Supporting Figures (Fig. S16, S19) and new Supporting Movies (S9 and S14) and are briefly mentioned in the main text (p 16, end of final paragraph).

Minor issues.

1) The authors claim that their new model solves the inconsistencies of previous models and experimental data. Nevertheless, in its present form, the manuscript does not provide sufficient details and clarity on which are these points of inconsistency from the previous literature. This is a critical aspect

that needs more explanation, the reader should understand better the facts that drive the inconsistencies of the previous models and experiments.

Response and changes made: We are sorry that we have been unsuccessful in effectively conveying this information. We feel that this issue is strongly related to the lack of clarity associated with the major issue 1 pointed out by this reviewer (see above). We have now made extensive efforts to clarify these points. This includes modifications in: i. the Introduction (on p 2, 2nd paragraph and p3, 1st paragraph), ii. the Results (p 11-12) and iii. the Discussion (p 19, 1st paragraph).

*2) In their structural analysis, the authors omitted the possibility found by other structural studies (<https://doi.org/10.1021/acs.jpcc.0c10004>) (Mauro Lorenzo Mugnai D. Thirumalai. *J Phys Chem B* 2021) that, instead of a single pathway, several gateways are present for the exit of Pi. If such circumstance is included, it is possible that the predictions of their model for the functioning of the myosin could change. Please discuss this issue.*

Response and changes made: We thank the referee for pointing out the recent simulation study of myosin VI by Mugnai and Thirumalai, which we now refer to in the revised version (p5, first paragraph). As these authors wrote, although there are multiple possible pathways for Pi-release, the backdoor is the dominant one. Given that this is the case, and in line with the earlier study by Cecchini and Karplus² and experimental information (e.g. ³), we consider the back door route to be the most accessible and hence most relevant for our detailed analysis. Given that other possibilities for Pi-escape are less frequent, and given that our measurements also allude to Pi-binding on the myosin surface we do not expect the presence of such route to modify the results.

3) The manuscript places high relevance to the delay of the release of Pi from the myosin into the solution. Nevertheless, the manuscript does not provide information on how long this delay is to be expected. Please detail this point.

Response: This is a highly relevant point. What we can demonstrate by molecular modelling is the existence of a secondary site in myosin II, similar to the one previously demonstrated in myosin VI. We also found evidence for a reasonably high affinity for Pi at this site. Furthermore, we present evidence for tertiary surface sites for Pi-binding.

Unfortunately, however, the actual off-rate from the secondary site is currently neither accessible to experimental measurements nor to reliable estimation by molecular modelling. In the latter case, there are simply too many complexities that cannot be faithfully included in the modelling as now specified on p 5, final 5 lines just before Fig. 1.

Overall, the experimental data of Muretta et al.⁴ and previously White et al.⁵ make it necessary to account for a delay of ~10-20 ms before Pi appears in solution after being released from the active site. Some of this delay (< 7 ms) can possibly be accounted for by binding on the myosin surface, outside the Pi-release tunnel. This possibility is now more strongly emphasized in the

main text, e.g. p 8, first paragraph. However, we still believe that strong binding at the secondary site accounts for most of the delay. This is supported by the actual experimental observation of bound Pi at this secondary site in myosin VI³ and our corroboration of the strong binding in myosin II using modelling. Our mechanokinetic modelling data, whose success heavily relies on the assumption of Pi-binding outside the active site, is further independent support of the idea of prolonged Pi-binding at secondary (and tertiary) sites as it allows reconciliation of a range of previously conflicting views.

Changes made: We address this issue by several changes to the new version of the manuscript. First, we emphasize more strongly (p 8 first paragraph and, Discussion, first paragraph) that both binding to the secondary site and to the tertiary surface sites are likely to contribute to the delay. Second, we explicitly acknowledge that the off-rate of Pi from the secondary site is presently not known (end of p 5, just before Fig. 1) and we emphasize the importance of devising an approach to directly measure this delay in future work (Discussion, p 19, end of first paragraph).

4) As the manuscript is structured, there is a lot of important information placed in the figure captions that are missing from the main text. To facilitate the reading, the distribution of the information should be more equilibrated.

Response and changes made: We agree that the figure legends are a bit extensive. We have now omitted some text from the legends (e.g. Fig. 4). Whereas we have not directly transferred text from the legends to the main text we have expanded the main text with further details and clarifications throughout as specified in response to other comments.

Response to Reviewer #2 (Please note, page numbers etc. refer to the version of the manuscript **with** track changes)

A new model is proposed to explain the relationship between phosphate release and force generation in skeletal muscle myosin, which is a crucial for understanding energy transduction in myosins and relevant to other mechanoenzymes. The key feature of this model is that Pi can bind to an alternate site on myosin and thus Pi-release occurs in two phases – Pi leaving the active site and Pi release from 2ndary binding site. This model is similar to what was proposed by Sweeney and Houdusse based on x-ray crystallography studies of myosin VI. The authors provide experimental support for their hypothesis using single molecule binding events with fluorescent ATP in the presence of varying Pi concentrations. They also use high-speed atomic force microscopy to demonstrate the lever arm rotation can occur with ADP in the active site. In addition, a mechanokinetic model was developed to test their hypothesis. In their model, the alternate Pi binding site does not alter the myosin power stroke making the single molecule mechanical experiments independent of free Pi concentrations. Pi binding to the second site does reduce isometric force without altering contractile velocity.

This is a very interesting study that provides new insight into a long standing question in the molecular motors field. The new model is very exciting and mostly well explained in the article and the supplementary information.

Response: We are grateful for this positive assessment.

However, the experimental support for the secondary Pi binding sites has some weaknesses. Also, in my opinion the high speed AFM results are over interpreted.

(Figure 1) The molecular modeling is very impressive and convincingly demonstrates the possibility of a secondary Pi binding site in myosin II. (Figure 2)

Response: Thanks for the positive assessment of the molecular model. We address the other issues below.

Specific comments on single molecule ATP binding events in the presence of Pi. The authors observed a reduction in the number of fluorescence spots with increasing Pi, suggesting competition between Pi and the fluorescent ATP for binding to myosin. This could be because they both compete for the active site. It has been found previously that Pi can compete with ATP for the active site (Amrute-Nayak et al. 2008)(Debold et al. 2011).

The authors go on to demonstrate several different dwell times whose amplitudes were decreased by Pi, consistent with different binding sites for Pi that alter the period of time the fluorescent nucleotide is bound. However, another interpretation of these results is that the fluorescent ATP can bind specifically and go through an ATPase cycle, non-specifically and rapidly detach, or bind to the active site but detach prematurely before going through an ATPase cycle. The myosin can also be in different conformations that could explain the different ATP binding/dissociation rates. The overall number of spots is dramatically reduced because of the ability of Pi to compete for the active site, which explains the most striking result. (Figure 3) The authors use ADP.vanadate to block the active site and address my concern above about competition for the active site. They find that the number of spots is reduced dramatically, which is expected since all or most active sites are blocked. The binding events that they do observe are similar to Figure 2. However, vanadate is not perfect and there could be some active sites that are free. The authors could verify experimentally that under the conditions of the experiment all active sites contain ADP.vanadate (blocked) using ATPase measurements. Overall, the number of spots is reduced because most of the active sites are blocked but those that are open behave the same as Figure 2. Pi can still compete for the active site as stated above.

Response: First, our results are consistent with previous data (e.g. ^{6,7}) that there is competition for the active site between Pi and ATP. Thus, because we have previously⁸ provided strong evidence that the slow phase of Alexa-ATP binding is due to its binding to the active site, inhibition of this slow phase by Pi is consistent with binding of Pi to the active site as in references ^{6,7}.

However, recently⁸ we also provided equally strong evidence that the fast phases of Alexa-ATP binding reflect non-specific binding outside the active site of myosin. The detail of this evidence is given in reference ⁸, but, in summary, the main arguments are as follows:

1. When ADP was locked by vanadate to the active site of myosin S1 (thus blocking the active site), the amplitude of the slow phase of Alexa-ATP binding to myosin subfragment 1 (corresponding to the myosin ATPase rate $\sim 0.05 \text{ s}^{-1}$) was markedly reduced whereas the unexplained phase of Alexa-ATP binding (rate $\sim 0,3 \text{ s}^{-1}$) was not. Rather a faster phase emerged ($\sim 3.5 \text{ s}^{-1}$). This suggests that fast phases are attributed to Alexa-ATP binding outside the active site. Since we used myosin subfragment 1, in the previous study, this binding would be on the motor domain, rather than somewhere on the subfragment 2 of HMM.

2. Nevertheless, in relation to point 1, we also agree with the reviewer that vanadate is unlikely to block the active sites to 100%. We therefore previously ⁸ performed additional experiments using fluorescent Alexa-ADP to confirm that the active site was actually occupied by Alexa-ADP and vanadate. This was indicated by extremely long fluorescence dwell times, rate limited by fluorescence photobleaching. Importantly, in those experiments, binding events corresponding to the fast phases occurred on top of the long-time binding of the vanadate-captured Alexa-ADP, temporarily doubling the fluorescence intensity (Fig. S11-12 in reference ⁸; reproduced in Figures R7-R8 below). Moreover, the rate constants associated with the fast (top) events (Fig. R7) were similar to those of the fast Alexa-ATP binding events found otherwise (Fig. R8). Importantly, the mentioned events with double intensity were not observed in the absence of blocking of the active site with Alexa-ADP and vanadate. Thus, the results described in this paragraph virtually proves that Alexa-ATP bound non-specifically outside the active site when the active site was blocked by Alexa-ADP.

As supplementary evidence to the very strong evidence in point 2:

3. We performed control experiments to exclude non-specific binding of the Alexa-moiety alone of Alexa-ATP to HMM (Fig S6 in reference ⁸)

4. We found several (6-12; 95% CI) Alexa-ATP binding sites on HMM using equilibrium dialysis (Fig S14 in reference ⁸).

5. Bioinformatics analysis suggested the presence of several ATP binding sites outside the active site of myosin subfragment 1 (Fig 5 in reference ⁸).

In view of the above, we feel that we cannot interpret reduced amplitudes of the fast phases of the Alexa-ATP binding in any other way than as inhibition of the Alexa-ATP binding outside the active site due to competition with Pi. It is important in this context to note that ionic strength was kept constant in our experiments with varied Pi excluding this basis for reduced Alexa-ATP binding with increased [Pi].

We have changed the manuscript at the end of p. 7 to emphasize more strongly that we are highly convinced about our interpretation of the Pi-effects on the Alexa-ATP binding. We have also indicated the basis for that conviction by now adding “convincingly” to the sentence: “Recent experimental results,

comparing isolated HMM and myosin subfragment 1 motor domains, **convincingly** attributed the fast phases to binding to the myosin head outside the active site ⁸. “

Summary of changes made: We have now more explicitly emphasized how the present results build on the convincing evidence that we recently reported for auxiliary binding sites for Alexa-ATP outside the active site of myosin on p 7, final 5 lines. Of relevance in this connection, we now also cite a recent paper⁹ showing evidence for such Auxiliary ATP binding sites outside the active site in another molecular motor.

Fig. R7. Reproduction of Fig. S11 of paper ⁸. Representative time traces of Alexa-ATP binding to fluorescent S1*aD*Vi complex hotspots consistent with unspecific Alexa-ATP binding to myosin outside the active

site. F(S1*aD·Vi): long fluorescent dwell times representing bleaching-limited fluorescence of S1*aD·Vi complex. F(aT|S1 S1*aD·Vi): short fluorescence dwell times on top of longer one, attributed to unspecific interaction of Alexa-ATP with S1 myosin outside the active site. For further details, please see reference ⁸. S1: myosin subfragment 1. aD: Alexa647-ADP. Vi: Vanadate.

Fig. R8. Reproduction of Fig. S12 of paper ⁸. **a** Triplicate of cumulative frequency distributions of Alexa-nucleotide dwell-time events (see Figure 1 above) on myosin subfragment 1 (S1) surface hotspots. Data from 36 (#1), 25 (#2), 29 (#3) S1*aD·Vi hotspots where the nucleotide pocket was blocked by fluorescent Alexa-ADP and Vanadate. The data were fitted by double exponential functions (solid lines). **b** Amplitudes and rate constants obtained from fittings to data to the left are comparable to the data when non-fluorescent ATP was used to block S1 active site ⁸. Notably basal ATPase activity ($k_{ATPase1} \sim 0.05 s^{-1}$) is here not detected, consistent with the fact that we analyzed only events on top of prolonged fluorescent signals attributed to the S1*aD·Vi complex (i.e. S1 with blocked active site). Error estimates refer to 95 % confidence intervals obtained in the fits. Temperature: 23 °C. For further details, please see ⁸.

(Figure 4) In the kinetic model is unclear how the presence of excess Pi alters specific reaction steps. This is important because the data demonstrate that the isometric force is sensitive to Pi but the shortening velocity is not. They also demonstrate that Pi does not alter the power stroke. This is exciting because the model agrees with experimental data but it could be explained better in the text the key steps that are Pi-sensitive and how changes in specific reaction steps lead to the outputs (Figure 4, b-d).

Response and changes made: We have now added new text in the end of p12 and beginning of p13, just above Fig. 4 and hope that this fully addresses the raised issue.

(Figure 6) The high speed AFM was performed to demonstrate a power stroke can still occur with only ADP in the active site. Since the model proposed required Pi-release from the active site first, the power stroke should occur with ADP in the active site. It is known that there is an additional swing of the lever arm with ADP release, although typically not detectable in skeletal muscle myosin, which can complicate the interpretation of experiments. The authors demonstrate mainly one conformation of the lever arm in ADP and another conformation in ADP-vanadate. There are two conformation in the presence of the drug blebbistatin which traps ADP in the active site. This is quite interesting and suggests the drug allows the lever arm to adopt two different conformations. However, it is unclear if the results demonstrate the movement of the lever arm from the pre to the post-power stroke conformation. For example, is it clear that myosin is remaining in a strongly bound conformation during the transition? The traces look like in all three conditions the lever arm can adopt a pre or post-power stroke. The data emphasizes the different conformations of the lever arm that are possible in each condition but does not emphasize specific events that demonstrate a transition from the pre- to post-power stroke state.

Response: The reviewer is correct. Due to time resolution limit, we were not able to observe the gradual transition from the pre- to post-power stroke. In order to do that, we would need an even higher frame rate (eg. 50 frames per second (fps) or 20 ms/frame). However, the spatial resolution at these conditions would drop significantly for observation of highly dynamic myosin molecules. The hs-AFM that we used is one of the fastest reported in the literature, but the technology is still advancing.

In some movies there are frames, where the change in the lever arm position from the pre-power-stroke to the post-power-stroke state can be readily observed. However, because of the limit in time resolution, the image frames represent the two stable states before and after the power-stroke or its reversal. This is why we used our novel experimental approach with each HMM head bound to two different nearly parallel actin filaments simultaneously¹. This ensured that we track lever arm dynamics in a given HMM molecule through the successive hs-AFM frames. To improve the visibility of the myosin heads and the lever arms, we have now applied a 3D-surface plot isoline filter (imageJ, v.1.47) to the movies and figures (new Movies S2-S7, Figs.S12-S14). These filtered hs-AFM movies were combined with the original hs-AFM movies for better correlation of the lever arm dynamics under the conditions evaluated in the study, where the orientation of the lever arm was highlighted in some cases to indicate how the angle was measured.

Despite our careful procedure as described above, we are aware that the exact lever arm orientation may be challenging to judge without ambiguity in some cases. It is therefore important to note that some clear changes in orientation between pre- and post-power stroke states exist, e.g. as indicated in Fig. R9 (Fig. S15). Several such changes can also be found through frame-by-frame inspection of the movies, particularly Movies S5, S6 and S7. Examples are seen during the time periods 1.2 s-1.4 s in Movie S6 and 2.40-2.55-2.70-2.85-3.0 s as well as 3.45-3.65 s in Movie S7 (see also hs-AFM Source dataset (Excel) for angle measurements and new Movies S15-S17). However,

particularly in Movies S5-S7, several more events can be observed by careful inspection frame by frame.

While we take the mentioned non-ambiguous events as strong support for our claims, one may of course, argue that the observed transitions are due to detachment from the post-power-stroke state followed by a recovery stroke in the detached head with transition into the pre-power-stroke state and subsequent reattachment to actin in the pre-power-stroke state conformation. However, in our experimental approach when each head is bound to two nearly parallel actin filaments simultaneously and in the absence of ATP the latter scenario is unlikely.

Summary of changes made: We have updated and expanded the supporting Movies directly cited in the text (Movies S2-S7) and combined them with 3D-surface plot isoline filtered versions of the movies to better visualize the myosin lever arm. We have also included additional movies S8-S17 for all studied conditions as well as a the hs-AFM Source Data file reporting x,y, z coordinates, lever arm angles and drift analysis. Some of these have contributed data to Fig. 6 as now indicated in the legend of that figure. Finally, we have modified the text to better guide the readers, e.g. with regard to the importance of observing movies frame by frame (p 15, end of final paragraph and p 16 as well as legends of Supporting Movies). We have now also more openly exposed the challenges in interpreting hs-AFM data (p 15, end of final paragraph and p 16).

Fig. R9. New Fig S15. Measurement of the lever arm position in the actin-myosin complex. **a** hs-AFM successive images of actin-myosin-ADP-vanadate complex with myosin heads bound to the two parallel actin filaments and **b** corresponding lever arm and heads orientations used to calculate the angle of the lever arm. The transition from the strong (0.15 s) to the weak (0.30-0.45 s) lever arm positions of the lower myosin head is indicated.

Response to Reviewer #3 (Please note, page numbers etc. refer to the version of the manuscript **with** track changes)

This manuscript details a combined experimental and modeling approach to determining the detailed kinetics of Pi release during the motor activity of myosin. The authors show that in certain isoforms of myosin, there is a "back door" binding site for the Pi. Using experiments and modeling, the authors determined the rate constants for binding of Pi, and showed that there are multiple rate constants, consistent with multiple binding sites. Using this analysis, the authors created a revised model for the power stroke kinetics of myosin that explain a number of aspects that could not be explained previously, most notably the Pi independence of the power stroke rate.

Overall, the results presented here provides a more detailed and quantitative model for the myosin-actin interaction. It can explain some aspects relevant to disease, such as why certain mutations are pathological. I am not an expert on the experimental approach, but it seemed fine. The modeling work was sound and supported the overall conclusions that were determined.

While this work will definitely impact the molecular motor community, its major weakness is that the results may not provide significant impact to fields outside this fairly narrow community.

Response: We thank the reviewer for the generally positive assessment of our manuscript. However, we are somewhat surprised by the claims regarding the “fairly narrow” molecular motors community. First, molecular motors are the basis for all movement of cells and organisms as well as intracellular transport and force-generation, something that is central overall in biology e.g. in cytoplasmic streaming of plants, multicellular animal locomotion (from jellyfish to humans), embryonal development, cytokinesis and key functions of the nervous system, the immune system etc. just to mention some examples. In several of these functions/events, myosin motors are central (as opposed to other molecular motors such as kinesin and dynein). Additionally, myosin motors play key roles in diseases not only of the heart and skeletal muscle, with associated on-going drug development as we discussed in the paper. The myosin motors also play important roles in cancer metastasis and in cellular invasion by malaria plasmodia with potential future drug developments.

Generally, the wide importance of the myosin motors in both biology and medicine is readily appreciated by browsing through the search results on PubMed using the search terms “Myosin AND Review” with more than 4000 hits (i.e. of Review papers only). Five references that we found in such a search from the last few years include ¹⁰⁻¹⁴. Together with the references we cite in the paper, these citations exemplify the importance of the myosin (particularly myosin II) motor system throughout diverse fields of science. To further emphasize this, we have now added one of the above references to those in the paper (in the Introduction on p 2 line 3).

Our analysis in the present paper suggests that the Pi-release is likely to be related to the force-generation in a similar way in most myosins. On a background of the above account, it would be difficult to see this issue as being of interest only for a “fairly narrow community”.

Finally, and additionally, as pointed out in several places in the manuscript, e.g. at the end of the Abstract and the end of the Discussion, our results have potential impact outside the molecular motors field.

Response to Reviewer #4 (Please note, page numbers etc. refer to the version of the manuscript with track changes)

I congratulate the authors for this highly interesting manuscript. The idea of unifying conflicting perspectives of power stroke and Pi release coupling by implementing a second binding site of the cleaved Pi in the kinetic scheme is ingenious and strongly supported in my feeling by the high-speed AFM structures of myosin presented in the paper.

Response: We are grateful to the reviewer for this positive assessment.

As I was requested by the editor to particularly review the kinetic models in the manuscript, I have only one main comment addressed to the author(s) performing the model computation. That comment is, however, really important because addressing it will greatly increase the significance of the models in terms of both, reliability of rate constants and consistency with thermodynamics.

The models in the manuscript consisting of branched pathways do not account for thermodynamics. When models implement different pathways leading from the same initial state before branching to the same final state after the branches, the values of rate and equilibrium constants within the branched part of the model cannot freely selected, some of them have to be adjusted accordingly by the principle of detailed balance to fulfil thermodynamics. I am almost sure that the expert performing the simulations is aware of this. However, in this regard, the transition in Fig. 4a leading from the AMDP'H to the AM/AMD state is superficial and inconsistent with thermodynamic principles. If it would not exist and power stroke transitions and second release steps of Pi are the same in the two diverging pathways as in this manuscript, the model would be consistent.

Similarly, in Fig S8a and b, the merging transitions from different initial states to the same final ATP state (designated with the same rate constant and computed by using the same values for k_{det} in the computations) are also inconsistent with thermodynamics. There should be only one transition in each model leading to the ATP state, i.e., the transition from the ADP state. The other transitions designed by this merging arrow are superficial and already covered by the forward transitions leading to the ADP state in the model and their inclusion contradicts thermodynamics. Thus, I suggest correcting the model schemes in Fig. 4 and Fig. S8 by deleting the superficial transitions.

Response and changes made: We thank the reviewer for these comments that are highly relevant and important. We are sorry that we did not treat this issue with sufficient rigor previously, i.e. with regard to fulfilling detailed balance

conditions. As the reviewer indicates, we did tacitly assume that detailed balance conditions are fulfilled but we did not clarify how, because we thought of different possibilities. Our ideas in this regard differ from the suggestion put forward by the reviewer. Whereas the latter suggestion would make the model thermodynamically consistent, it would probably not maintain the unique properties of the model that allows it to reconcile findings such as the lack of effects of $[Pi]$ on velocity and the appearance of Pi in solution after the power-stroke with a time constant of about 10 ms. Thus, if the transition to the AM/AMD state would occur only from the AMDH state and not also from the AMDPH state, the Pi -release step would limit the maximum shortening velocity, leading to appreciably lower velocity than observed experimentally as well as a strong dependence of velocity on $[Pi]$, which is not seen.

Fig. R10. New Figure 4a in paper. For details, please see legend in paper.

In our explicit modification of the model, to formally ensure thermodynamic consistency, we rely on the assumption, as for the power-stroke, that Pi -binding at the secondary site does not affect rates of inter-state transitions along the second and third rows in the scheme in Figure 4a (reproduced here as Fig. R10). Therefore, it is natural to assume that the equilibrium governed by K_c (and K_c' in Fig. S8) has similar kinetics also in the AM/AMD state (denoted AM below). This is also consistent with an open switch II conformation in the rigor state, suggested by previous studies¹⁵. In this scheme (Fig. R10), either the re-binding of ATP or some of the fast subsequent transitions¹⁶ associated with the recovery stroke and the ATP hydrolysis is believed to disrupt the secondary Pi -binding site by closure of switch II^{17, 15, 18}. The equilibrium constant of this transition ($MTP' \leftrightarrow MT$ in upper row) may be identical to K_c but the rate constants may be orders of magnitude faster (i.e. $\gg 100 \text{ s}^{-1}$). This is, however, only one out of several possible ways to maintain detailed balance. Alternatively, the fast Pi -release could occur later in the cycle for detached myosin or it could occur simultaneously with detachment from actin or even from the rigor state provided that this can be made consistent with other important features of the model. Presently we favour the idea presented in Fig. R10 (new Fig. 4a in the paper) but we are open for other possibilities. The ideas outlined above may be readily extended to the situation in Fig. S8b. Importantly, the present formally correct kinetic schemes would

not give different quantitative results from those that we derived using the previous versions. This follows from the facts i. The AMDH/AMDHP' >AM/AMP' transitions are assumed irreversible (at negligible [MgADP] as assumed here) and ii. No transition between actomyosin states is affected by binding of Pi to the secondary site and iii. The release of Pi from the secondary site is very fast as soon as the myosin cross-bridge has detached from actin.

As an important consequence of point ii, all the transitions that rate limit cross-bridge detachment at low strain (from AMDH and AMDPH states to the M and MP' states), are assumed to occur at the same rate whether Pi is bound to the second site or not.

*Consequently, for making models correct, delete in the model equations the superficial -kdet*aXX-terms in the differential equations (where aXX are other states than the ADP state), re-run the corrected models and re-plot the new values obtained by the simulations in the figure templates.*

Response: We have now addressed this issue, in a slightly different way than proposed, as described in detail above.

Minor

comments:

Fig. S8 caption: "On the assumption that $k_{det}=0$ the isometric steady-state force as a function of $[Pi]$ is given by the expression in the box (with constants $c1$ and $c2$ given to the left). This expression gives a force- $[Pi]$ relationship that is negligibly different from that obtained by numerical solution of the differential equations with $k_{det}=5$ s⁻¹ (plotted in main Fig. 4c)." Does this also account for the corrected model?

Response: Yes it does, provided that K_C in the equation is substituted by K_C' K_C'' . This is now mentioned in the legend of Fig. S8.

Generally, the literature is well cited and balanced in the manuscript. I would like to point to studies of Lionne et al (1995) FEBS Letters with myofibrils and He et al (1997) J Physiol with muscle fibres that have demonstrated that there is no burst of free Pi after initiating the cross-bridge ATPase cycle and muscle fibres. The surprising absence of free Pi bursts in these studies provides straightforward evidence for delayed release of phosphate by working cross-bridges addressed in the manuscript.

Response: We thank the reviewer for suggesting these papers which are now considered towards the end of the first paragraph of the Discussion.

*Yours, sincerely
Robert Stehle, PhD*

References

- 1 Matusovsky, O. S. *et al.* Millisecond Conformational Dynamics of Skeletal Myosin II Power Stroke Studied by High-Speed Atomic Force Microscopy. *ACS nano* **15**, 2229-2239, doi:10.1021/acsnano.0c06820 (2021).

- 2 Cecchini, M., Alexeev, Y. & Karplus, M. Pi release from myosin: a simulation analysis of possible pathways. *Structure* **18**, 458-470, doi:10.1016/j.str.2010.01.014 (2010).
- 3 Llinas, P. *et al.* How actin initiates the motor activity of Myosin. *Dev Cell* **33**, 401-412, doi:10.1016/j.devcel.2015.03.025 (2015).
- 4 Muretta, J. M., Rohde, J. A., Johnsrud, D. O., Cornea, S. & Thomas, D. D. Direct real-time detection of the structural and biochemical events in the myosin power stroke. *Proc. Natl. Acad. Sci. U. S. A.* **112**, 14272-14277, doi:10.1073/pnas.1514859112 (2015).
- 5 White, H. D., Belknap, B. & Webb, M. R. Kinetics of nucleoside triphosphate cleavage and phosphate release steps by associated rabbit skeletal actomyosin, measured using a novel fluorescent probe for phosphate. *Biochemistry*. **36**, 11828-11836, doi:10.1021/bi970540h (1997).
- 6 Amrute-Nayak, M., Antognozzi, M., Scholz, T., Kojima, H. & Brenner, B. Inorganic phosphate binds to the empty nucleotide binding pocket of conventional myosin II. *J. Biol. Chem.* **283**, 3773-3781 (2008).
- 7 Debold, E. P., Turner, M. A., Stout, J. C. & Walcott, S. Phosphate enhances myosin-powered actin filament velocity under acidic conditions in a motility assay. *Am J Physiol Regul Integr Comp Physiol* **300**, R1401-1408, doi:10.1152/ajpregu.00772.2010 (2011).
- 8 Usaj, M., Moretto, L., Vemula, V., Salhotra, A. & Mansson, A. Single molecule turnover of fluorescent ATP by myosin and actomyosin unveil elusive enzymatic mechanisms. *Commun Biol* **4**, 64, doi:10.1038/s42003-020-01574-0 (2021).
- 9 Zananiri, R. *et al.* Auxiliary ATP binding sites support DNA unwinding by RecBCD. *Nature communications* **13**, 1806, doi:10.1038/s41467-022-29387-1 (2022).
- 10 Houdusse, A. & Titus, M. A. The many roles of myosins in filopodia, microvilli and stereocilia. *Curr. Biol.* **31**, R586-R602, doi:10.1016/j.cub.2021.04.005 (2021).
- 11 Trivedi, D. V., Nag, S., Spudich, A., Ruppel, K. M. & Spudich, J. A. The Myosin Family of Mechanoenzymes: From Mechanisms to Therapeutic Approaches. *Annu. Rev. Biochem.* **89**, 667-693, doi:10.1146/annurev-biochem-011520-105234 (2020).
- 12 Heissler, S. M. & Sellers, J. R. Kinetic Adaptations of Myosins for Their Diverse Cellular Functions. *Traffic* **17**, 839-859, doi:10.1111/tra.12388 (2016).
- 13 Garrido-Casado, M., Asensio-Juarez, G. & Vicente-Manzanares, M. Nonmuscle Myosin II Regulation Directs Its Multiple Roles in Cell Migration and Division. *Annu. Rev. Cell Dev. Biol.* **37**, 285-310, doi:10.1146/annurev-cellbio-042721-105528 (2021).
- 14 Day, S. M., Tardiff, J. C. & Ostap, E. M. Myosin modulators: emerging approaches for the treatment of cardiomyopathies and heart failure. *J. Clin. Invest.* **132**, doi:10.1172/JCI148557 (2022).
- 15 Reubold, T. F., Eschenburg, S., Becker, A., Kull, F. J. & Manstein, D. J. A structural model for actin-induced nucleotide release in myosin. *Nat. Struct. Biol.* **10**, 826-830, doi:10.1038/nsb987 (2003).
- 16 Johnson, K. A. & Taylor, E. W. Intermediate states of subfragment 1 and actosubfragment 1 ATPase: reevaluation of the mechanism. *Biochemistry*. **17**, 3432-3442. (1978).
- 17 Furch, M., Fujita-Becker, S., Geeves, M. A., Holmes, K. C. & Manstein, D. J. Role of the salt-bridge between switch-1 and switch-2 of Dictyostelium myosin. *J. Mol. Biol.* **290**, 797-809, doi:10.1006/jmbi.1999.2921 (1999).

- 18 Koppole, S., Smith, J. C. & Fischer, S. The structural coupling between ATPase activation and recovery stroke in the myosin II motor. *Structure* **15**, 825-837, doi:10.1016/j.str.2007.06.008 (2007).

REVIEWERS' COMMENTS

Reviewer #1 (Remarks to the Author):

I congratulate the authors. The issues that I raised before have been well addressed: (i) Concerning the hs-afm data, the lack of statistical analysis and the lack of clear distinction in the analysis between the experimental and imaging conditions have been well addressed. Moreover, the reviewer acknowledges that the new version provides more movies and is a sufficient hs-afm dataset to support the statistical analysis. (ii) The new version is better structured and easy to read. (iii) The new version correctly frames the findings in the previous literature. (iv) The new version signals that the measurement lacks experimental assessment of the rate constants. The manuscript is well balanced at this point. Its content is relevant, especially for the field of molecular motors, yet implications go beyond the motors as they reach the mechanisms of enzyme functioning. Considering the improvement the authors implemented, I recommend the publication of the manuscript at Nature Communications.

Reviewer #2 (Remarks to the Author):

The authors have done an excellent job of addressing the reviewer's comments. The additional controls with the single molecule experiments show convincingly that there are secondary Pi binding sites on the motor domain. The new AFM data with HMM improve the resolution of the pre- and post-power stroke conformations. The authors have improved their description of the mechanokinetic model and made it more accessible to a broad readership. Overall, I find that the manuscript makes an excellent contribution to the literature on myosins, muscle contraction, and the general function of mechanoenzymes. However, I still question how some of the stronger statements are presented by the authors.

Please consider addressing the following comments before publication.

1) Abstract. This statement in the abstract may be too strong. "The model is further supported by high-speed atomic force microscopy data that directly visualize power strokes by single myosin II molecules without Pi at the active site." I would argue that the power stroke was not directly visualized in this study. I would argue that snapshots of the pre- and post-power stroke conformation were visualized at different time points. I think this could be important because one view is that myosin can adopt different conformations in any state (e.g. pre and post-power stroke and open and closed actin binding cleft) while the rate/equilibrium constants govern the most probable conformational pathway.

2) In the first paragraph of the discussion the following statement is made. "Our results corroborate the idea that Pi-release from the active site precedes and triggers the power-stroke but that Pi does not appear in solution until after a delay due to binding to secondary site(s) in the Pi-release tunnel (Fig. 1c)."

What is the experimental evidence for Pi-release preceding the power stroke? I agree that there is evidence for secondary Pi binding sites and there is evidence that the lever arm can adopt the pre- and post-power stroke conformations in the presence of ADP. However, the evidence of the timing of Pi-release and the power stroke comes from the kinetic modeling and not directly from experimental results. The authors nicely describe the previous results in the literature (crystallography and FRET) and came up with a model that can comprehensively explain most/all of the data in the literature. Although parts of this model were proposed by Sweeney and Houdusse based on crystallography, the current paper incorporates the model into a mechanokinetic model which is very impressive. However, I don't feel the paper provides definitive experimental evidence for the timing of Pi release and the power stroke. This is a long standing question and other experiments have reported opposing views, such as the high speed optical trapping data (Woody et al.) which strongly favors the power stroke preceding phosphate release. Thus, the authors may want to temper their statements about the order of events associated with Pi release and the power stroke.

Reviewer #4 (Remarks to the Author):

The authors have addressed my comments and revised the model for reaching its consistency with the principles of thermodynamics. Though the authors chose a different way to reach this goal than I suggested, I am very satisfied with the revised model and the current version of the manuscript.

Dear Reviewers,

We are grateful for yet another careful and constructive review of our manuscript that includes further helpful suggestions for changes. Please find below, our detailed point to point response to the reviewer comments with the original comments in italics. In addition, the key changes are indicated by **track changes** in a submitted version of the manuscript for review only. Changes in the manuscript indicated by “track-changes” are also due to editor suggestions and journal style requirements. Finally, we have again carefully gone through the paper with respect to errors and with a major aim to further improve clarity. The resulting corrections, due to our own new reading of the manuscript are also generally included among those indicated by track changes. Page numbers etc. refer to the version of the manuscript **with** track changes.

Reviewer #1

I congratulate the authors. The issues that I raised before have been well addressed: (i) Concerning the hs-afm data, the lack of statistical analysis and the lack of clear distinction in the analysis between the experimental and imaging conditions have been well addressed. Moreover, the reviewer acknowledges that the new version provides more movies and is a sufficient hs-afm dataset to support the statistical analysis. (ii) The new version is better structured and easy to read. (iii) The new version correctly frames the findings in the previous literature. (iv) The new version signals that the measurement lacks experimental assessment of the rate constants. The manuscript is well balanced at this point. Its content is relevant, especially for the field of molecular motors, yet implications go beyond the motors as they reach the mechanisms of enzyme functioning. Considering the improvement the authors implemented, I recommend the publication of the manuscript at Nature Communications.

Response: Thank you for the positive assessment.

Reviewer #2:

The authors have done an excellent job of addressing the reviewer's comments. The additional controls with the single molecule experiments show convincingly that there are secondary Pi binding sites on the motor domain. The new AFM data with HMM improve the resolution of the pre- and post-power stroke conformations. They authors have improved their description of the mechanokinetic model and made it more accessible to a broad readership. Overall, I find that the manuscript makes an excellent contribution to the literature on myosins, muscle contraction, and the general function of mechanoenzymes. However, I still question how some of the stronger statements are presented by the authors.

Response: Thank you for the positive assessment.

Please consider addressing the following comments before publication.

1) Abstract. This statement in the abstract may be too strong. “The model is further supported by high-speed atomic force microscopy data that directly

visualize power strokes by single myosin II molecules without Pi at the active site.” I would argue that the power stroke was not directly visualized in this study. I would argue that snapshots of the pre- and post-power stroke conformation were visualized at different time points. I think this could be important because one view is that myosin can adopt different conformations in any state (e.g. pre and post-power stroke and open and closed actin binding cleft) while the rate/equilibrium constants govern the most probable conformational pathway.

Response and changes made: The reviewer arguments are reasonable. We therefore change the statement as follows in the Abstract:

“The model is consistent with high-speed atomic force microscopy movies of single myosin II molecules without Pi at the active site, showing consecutive snapshots of pre- and post-power stroke conformations.”

2) *In the first paragraph of the discussion the following statement is made. “Our results corroborate the idea that Pi-release from the active site precedes and triggers the power-stroke but that Pi does not appear in solution until after a delay due to binding to secondary site(s) in the Pi-release tunnel (Fig. 1c).” What is the experimental evidence for Pi-release preceding the power stroke? I agree that there is evidence for secondary Pi binding sites and there is evidence that the lever arm can adopt the pre- and post-power stroke conformations in the presence of ADP. However, the evidence of the timing of Pi-release and the power stroke comes from the kinetic modeling and not directly from experimental results. The authors nicely describe the previous results in the literature (crystallography and FRET) and came up with a model that can comprehensively explain most/all of the data in the literature. Although parts of this model were proposed by Sweeney and Houdusse based on crystallography, the current paper incorporates the model into a mechanokinetic model which is very impressive. However, I don’t feel the paper provides definitive experimental evidence for the timing of Pi release and the power stroke. This is a long standing question and other experiments have reported opposing views, such as the high speed optical trapping data (Woody et al.) which strongly favors the power stroke preceding phosphate release. Thus, the authors may want to temper their statements about the order of events associated with Pi release and the power stroke.*

Response and changes made: First of all, we wish to emphasize that the idea of Pi-release from the active site before the power-stroke is possibly more strongly supported by our kinetic modelling than indicated in the arguments above. Thus, if the data of Woody et al. are interpreted in terms of our model (rather than in terms of models with single-step Pi-release from the active site, either before or after the power-stroke) those data (Woody et al) also accord with Pi-release *from the active site* before the powers-stroke. Furthermore, our hs-AFM data and previous optical tweezers data (Hwang et al., (2021), PNAS 118), indicating repeated pre-post-power-stroke transitions without Pi at the active site, also accord with that view. Nevertheless, the reviewer criticism points to a limitation of our model that cannot be ignored, i.e. no direct experimental measurement exists to obtain quantitative information of the rate

of Pi-release from the active site vs the rate of Pi-release from secondary and tertiary site. If the entire delay of Pi-appearance in solution could unequivocally be attributed to binding to the secondary and tertiary site (based on measurements), our model would be closer to conclusively proven. Clearly, we are not quite there yet. Importantly, however, we are well aware of that fact and have already emphasized it in the most recent version of the manuscript with the sentences “A limitation of the model is that it is not fully defined quantitatively. Thus, it would be desirable to devise a method to more exactly measure the Pi-off rate constants from all Pi binding sites outside the active site.” at the end of the first paragraph of p. 16.

However, we admit that it could be of value to emphasize this remaining limitation also in the beginning of the Discussion. We have now, therefore, added a second sentence (“Limitations...discussed below”) after the first sentence as follows:

“Our results corroborate the idea that Pi-release from the active site precedes and triggers the power-stroke but that Pi does not appear in solution until after a delay due to binding to secondary site(s) in the Pi-release tunnel (Fig. 1c). Limitations in the evidence for this conclusion are discussed below.”

Reviewer #4 (Remarks to the Author):

The authors have addressed my comments and revised the model for reaching its consistency with the principles of thermodynamics. Though the authors chose a different way to reach this goal than I suggested, I am very satisfied with the revised model and the current version of the manuscript.

Response: Thank you for the positive assessment.